# A conserved differentiation programme facilitates inhibitory neuron production in the developing mouse and human cerebellum

Jens Bager Christensen[1,2], Alex P. A. Donovan[1,3], Marzieh Moradi[1,2], Giada Vanacore[1,2], Mohab Helmy[1,2], Adam J. Reid[1], Jimmy Tsz Hang Lee[4], Omer Ali Bayraktar[4], Andrea H. Brand[1,3] and N. Sumru Bayin[1,2,*]

**ABSTRACT**

Understanding the molecular mechanisms driving lineage decisions and differentiation during development is challenging in complex systems with a diverse progenitor pool, such as the mammalian cerebellum. Importantly, how different transcription factors cooperate to generate neural diversity and the gene regulatory mechanisms that drive neuron production, especially during the late stages of cerebellum development, are poorly understood. We used single cell RNA-sequencing (scRNA-seq) to investigate the developmental trajectories of nestin-expressing progenitors (NEPs) in the neonatal mouse cerebellum. We identified FOXO1 as a key regulator of NEP-to-inhibitory neuron differentiation, acting directly downstream of ASCL1. Genome occupancy and functional experiments using primary NEP cultures showed that both ASCL1 and FOXO1 regulate neurogenesis genes during differentiation while independently regulating proliferation and survival, respectively. Furthermore, we demonstrated that WNT signalling promotes the transition from an ASCL1+ to a FOXO1+ cellular state. Finally, the role of WNT signalling in promoting neuron production via FOXO1 is conserved in primary human NEP cultures. By resolving how cerebellar inhibitory neurons differentiate, our findings could have implications for cerebellar disorders such as spinocerebellar ataxia, where these cells are overproduced.

**KEY WORDS: Neural development, Molecular layer interneurons, Differentiation, Forkhead box O1, Cerebellum, Neurospheres**

**INTRODUCTION**

How neural stem cells generate diverse cell types with intricate cytoarchitecture in the brain remains a fundamental yet unresolved question in developmental biology. While fate mapping and clonal analyses are instrumental in resolving how distinct lineages are generated during nervous system development, the molecular mechanisms that drive lineage decisions and differentiation remain poorly understood, especially outside the forebrain. Single-cell genomic technologies have enabled the capture of transient cell states during lineage commitments and differentiation, providing the

[1]Gurdon Institute, Cambridge University, Cambridge CB2 1QN, UK. [2]Department of Physiology, Development and Neuroscience, Cambridge University, Cambridge CB2 3EL, UK. [3]Department of Cell Biology, NYU Grossman Medical School, New York, NY 10016, USA. [4]Wellcome Sanger Institute, Hinxton CB10 1RQ, UK.

*Author for correspondence (nsb44@cam.ac.uk)

N.S.B., 0000-0003-4371-855X

opportunity to map the dynamics of the gene regulatory networks (GRNs) driving these processes.

The cerebellum is a key hindbrain structure, crucial for motor coordination, as well as cognitive and social behaviours (Caligiore et al., 2017; Fatemi et al., 2012; Koziol et al., 2014; Lackey et al., 2018). Having co-evolved with the neocortex (Herculano-Houzel, 2010), the cerebellum has significantly expanded in mammals, housing ~60% and ~80% of the neurons in the mouse and human brains, respectively (Azevedo et al., 2009; Gill and Sillitoe, 2019; Herculano-Houzel et al., 2006). Although the cerebellar primordium is established at embryonic day (E) 9 in mice (Erickson et al., 2025; Leto et al., 2016), the majority of its growth occurs postnatally. The postnatal cerebellar progenitors continue to proliferate and differentiate up to 2 weeks after birth in mice and 6 months in humans (Erickson et al., 2025; Leto et al., 2016; Rakic and Sidman, 1970). Due to this protracted development, the cerebellum is particularly vulnerable to stress and damage around birth. Indeed, cerebellar injury at birth is the second leading risk factor for autism spectrum disorders (Wang et al., 2014). Furthermore, increasing evidence suggests that spinocerebellar ataxias manifest as developmental abnormalities in various cerebellar cell types, including altered progenitor cell behaviour and postnatal inhibitory interneuron overproduction (Becker et al., 2009; Edamakanti et al., 2018; Luttik et al., 2022). Therefore, dissecting the molecular mechanisms that regulate postnatal cerebellum development is crucial to understanding disease pathophysiology and identifying potential therapeutic interventions.

The postnatal development of the cerebellum is orchestrated by the coordinated action of lineage-restricted progenitors derived from two molecularly distinct embryonic germinal zones: the rhombic lip and the ventricular zone (VZ) (Joyner and Bayin, 2022; Leto et al., 2016; Wizeman et al., 2019). Rhombic lip-derived granule cell progenitors (GCPs) proliferate in the external granule layer and differentiate into the excitatory granule cells that migrate inwards to the internal granule layer in an outside-in manner (Machold and Fishell, 2005; Wang et al., 2005; Wingate and Hatten, 1999). On the other hand, VZ-derived nestin-expressing progenitors (NEPs) consist of multiple subtypes and give rise to molecular layer inhibitory neurons, Bergmann glia and astrocytes (Bayin et al., 2021; Lee et al., 2005; Milosevic and Goldman, 2004; Parmigiani et al., 2015). The *Hopx*-expressing gliogenic NEPs (*Hopx*-NEPs) are further divided into two subtypes based on their locations: those in the lobule white matter (WM) generate astrocytes, while those in the Bergmann glia layer (BgL) give rise to both astrocytes and Bergmann glia (Bayin et al., 2021; Cerrato et al., 2018). The neurogenic NEPs that express the basic helix loop helix proneural transcription factor *Ascl1* (*Ascl1*-NEPs) reside in the lobule WM of the postnatal cerebellum. The *Ascl1*-NEPs proliferate in the WM and generate the molecular layer inhibitory interneurons, basket and stellate cells (Leto et al., 2009). Once born in the lobule WM, the inhibitory neurons migrate radially to the

molecular layer in an inside-out manner (Leto et al., 2006, 2009, 2016; Sudarov et al., 2011; Zhang and Goldman, 1996). Furthermore, lineage-tracing studies have demonstrated that some NEPs in the WM are bipotent, generating both astrocytes and interneurons during early postnatal development (Bayin et al., 2021; Cerrato et al., 2018). Finally, a small subset of GCPs also express the NEP markers *Sox2* and *Nes* (Selvadurai et al., 2020; Vanner et al., 2014). While the molecular mechanisms that regulate GCP proliferation and differentiation have been extensively studied due to their role in some medulloblastomas (Cheng et al., 2020; Dahmane and Altaba, 1999; Gold et al., 2024; Miyata et al., 1999), the molecular mechanisms that govern NEP subpopulations remain largely unknown.

Cellular identities and the differentiation programmes required to achieve them are established by the combinatorial and context-dependent action of pleiotropic transcription factors. A recent analysis revealed that multiple transcription factors cooperate to mediate GABAergic differentiation in cerebral interneurons, highlighting the importance of transcription factor networks in development (Catta-Preta et al., 2024). However, the GRNs that drive molecular layer inhibitory neuron production in the cerebellum are not well understood, although ASCL1 is required for their differentiation (Grimaldi et al., 2009; Sudarov et al., 2011). ASCL1 is a pioneer proneural transcription factor that drives neurogenesis (Castro et al., 2006; Raposo et al., 2015; Woods et al., 2022) during development (Casarosa et al., 1999) and in direct reprogramming (Vierbuchen et al., 2010). A candidate family of proteins that could cooperate with ASCL1 is the forkhead-box transcription factor family subgroup O (FOXO) proteins. They play a crucial role in stem cell homoeostasis (Paik et al., 2009), cellular metabolism (Bastie et al., 2005; Puigserver et al., 2003) and apoptosis (Brunet et al., 2004; Papadia et al., 2008; Yuan et al., 2008). While some studies show that FOXO genes are important for neural stem cell homeostasis and promote stemness by inhibiting differentiation (Kim et al., 2015; Paik et al., 2009; Webb et al., 2013), others have suggested that they also regulate autophagy, migration and maturation (De La Torre-Ubieta et al., 2010; Huynh et al., 2011; Schäffner et al., 2018) in developing neurons. Finally, various ChIP-seq data suggest that ASCL1 and FOXO motifs co-occur in neural and brain tumour stem cells, suggesting overlapping roles (Park et al., 2017; Webb et al., 2013). While previous studies have shown that FOXO3 is a negative regulator of ASCL1-dependent neurogenesis (Webb et al., 2013), the functional relationship between FOXO proteins and ASCL1 remains largely unknown.

In this study, we used scRNA-seq, primary mouse and human NEP cultures, genetic induced fate mapping and conditional knockout (CKO) mice to investigate the molecular drivers of NEP-to-inhibitory neuron differentiation in the developing cerebellum. We demonstrate that, during *in vitro* differentiation of primary cerebellar neurosphere cultures, ASCL1 and WNT signalling upregulates FOXO1, which in turn promotes neural differentiation. Analysis of genome occupancy showed that ASCL1 and FOXO1 regulate a common set of neurogenic genes but also exhibit temporally divergent functions; ASCL1 promotes the proliferation of *Ascl1*-NEPs, whereas FOXO1 supports cell survival during *in vitro* differentiation. Finally, we demonstrate that the molecular mechanisms governing NEP-to-inhibitory neuron differentiation in mouse NEPs are conserved in primary human NEP cultures. Collectively, this study establishes core components of the GRN that drive NEP-to-inhibitory neuron differentiation in the developing mouse and human cerebellum.

## RESULTS

### Iterative subclustering reveals cellular states during NEP-to-inhibitory neuron differentiation during postnatal development

To identify key cellular states and the molecular mechanisms that drive NEP-to-inhibitory neuron differentiation during postnatal cerebellum development, we analysed previously generated scRNA-seq data of FACS-isolated CFP$^+$ cells from postnatal day (P) 1, 2, 3 and 5 *Nes-Cfp/+* mouse cerebella (Pakula et al., 2025) (Fig. 1A-D). Following quality control, 4872 cells were clustered into 11 clusters to identify distinct NEP subtypes and cellular states during the first 5 days after birth (Fig. 1E and Fig. S1A,B). Clusters were annotated using established lineage markers and highlighted the presence of the expected NEP subtypes and their immediate progenies, likely captured as a result of CFP perdurance. To this end, we identified *Hopx*-NEPs (clusters 0 and 6), *Ascl1*-NEPs (cluster 1) and their progenies: astrocytes (cluster 4, *Slc6a11*$^+$) and immature inhibitory neurons (cluster 3, *Pax2*$^+$), respectively (Bayin et al., 2021; Pakula et al., 2025).

Other *Nes*$^+$ populations, as previously observed (Pakula et al., 2025), were also detected. The *Nes-Cfp*$^+$ GCPs (clusters 2, 5 and 8, *Atoh1*$^+$/*Rbfox3*$^+$) represent a subpopulation of GCPs that express *Nes* and *Sox2* (Li et al., 2013; Vanner et al., 2014) (Fig. 1E and Fig. S1C). Whether these cells originate from the rhombic lip or the VZ, their physiological function and their relevance to previously reported *Nes*$^+$ and/or *Sox2*$^+$ GCPs remain to be determined. Other cell types that were observed include oligodendrocyte progenitors and oligodendrocytes (cluster 10, *Olig2*$^+$/*Pdgfra*$^+$), ependymal cells (cluster 7, *Foxj1*$^+$), mesenchymal cells (cluster 9, *Vtn*$^+$) and microglia (cluster 11, *Cx3cr1*$^+$), all of which were detected in various proportions across different postnatal days (Fig. 1E, Fig. S1A-C, Table S1). Based on previous fate-mapping and clonal analyses, these cells are not the primary progeny of the postnatal neurogenic and gliogenic NEPs. Therefore, clusters 7 and 9-11 were omitted from downstream analysis (Bayin et al., 2021; Cerrato et al., 2018; Fleming et al., 2013; Sudarov et al., 2011; Wojcinski et al., 2017). Finally, although a small proportion of Nes-CFP$^+$ cells was observed in the deep white matter via histological analysis (Bayin et al., 2021), these cells are likely under-represented in our dataset due to their rarity and were not detected as a separate cluster.

To increase the resolution of distinct NEP cellular states and to identify mechanisms of NEP-to-inhibitory neuron differentiation, we performed iterative subclustering. First, we subclustered all NEPs and their progeny from the initial dataset (clusters 0, 1, 3, 4 and 6, Fig. 1E). Clustering of 3357 NEPs showed increased molecular diversity highlighted by the presence of ten clusters (Fig. 1F and Fig. S1D-G, Table S1). BgL *Hopx*-NEP clusters N0, N1, N4 and N7 (*Gdf10*$^+$), WM *Hopx*-NEPs and *Ascl1*-NEPs [i.e. *Hopx*-NEPs (cluster N8) and *Ascl1*-NEPs (clusters N2 and N5)] were identified along with their immediate progenies: *Slc6a11*$^+$ astrocytes (cluster N10) and immature *Pax2*$^+$ inhibitory neurons (clusters N6 and N9). Finally, cluster N3 showed enrichment for *Pvalb* expression (Fig. S1F). Based on Allen brain atlas RNA *in situ* hybridisation data at P4, *Pvalb* is expressed in the cerebellar nuclei in the deep WM (Fig. S1H), perhaps representing some of the Nes-CFP$^+$ cells there. The inhibitory neurons of the cerebellar nuclei are generated directly from the ventricular zone between E10 and E13 (Leto et al., 2006; Sudarov et al., 2011) and are not the progeny of the postnatal neurogenic-NEPs. Thus, cluster N3 was excluded from further analyses.

We aimed to identify WM NEPs in the cerebellar cortex, which generates the molecular layer inhibitory neurons and astrocytes.

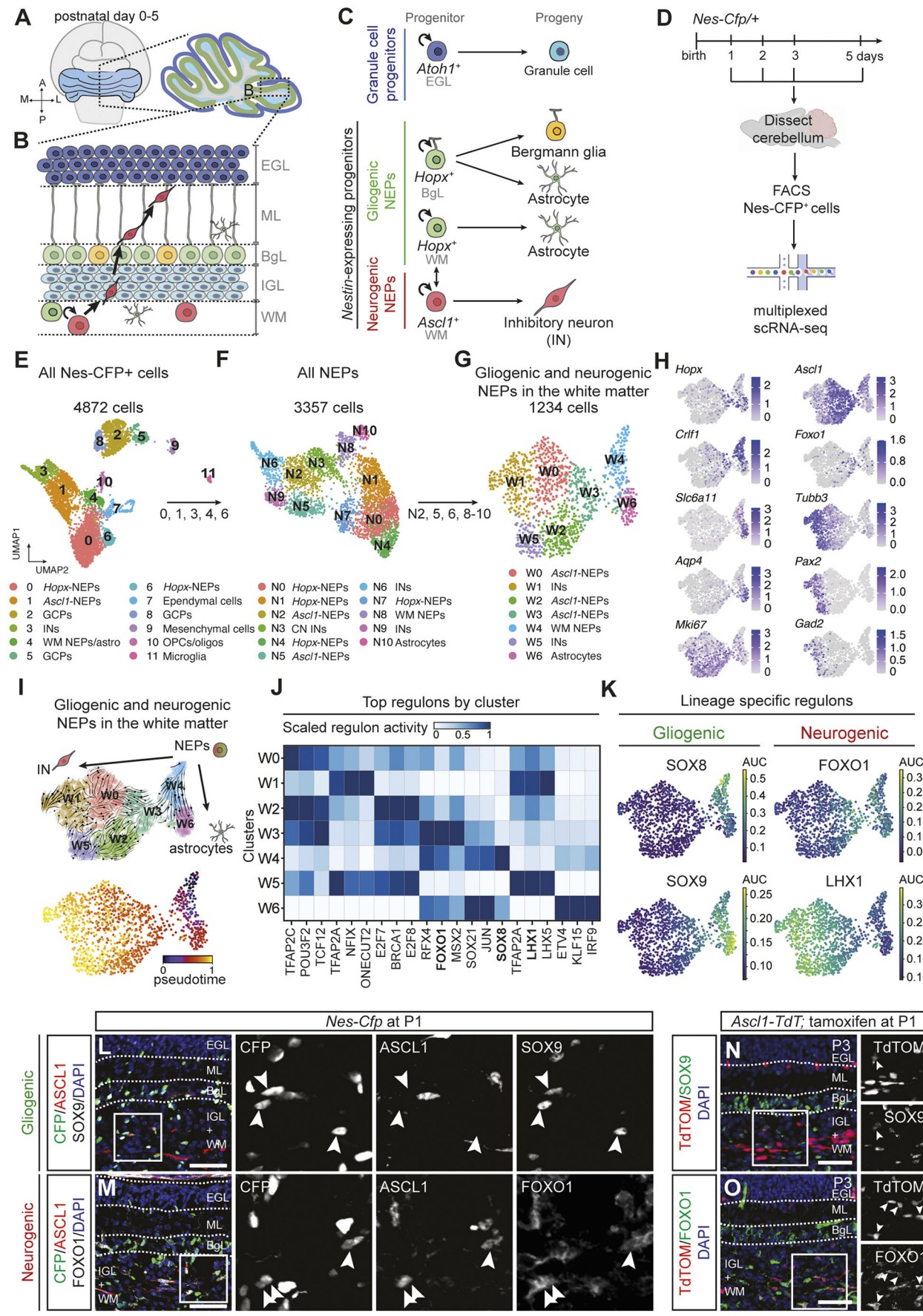

**Fig. 1.** See next page for legend.

**Fig. 1. scRNA-seq captures the molecular diversity of NEPs and reveals the regulatory mechanisms that drive the lineage decisions and differentiation during postnatal mouse cerebellum development.** (A-C) Schematics showing the postnatal cerebellar progenitors and their lineages. (D) Experimental design. (E-G) Uniform manifold approximation and projection (UMAP) visualisation of all Nes-CFP⁺ cells (4872 cells, E), all NEPs (3357 cells, F), and the gliogenic and neurogenic NEPs in the WM (1234 cells, G) labelled by cluster identity (Table S1 and Fig. S1A-J). (H) UMAP projection of normalised expression levels of known marker genes (gliogenic-NEPs, *Hopx*; astrocytes, *Slc6a11* and *Aqp4*; neurogenic-NEPs, *Ascl1*; immature inhibitory neurons, *Tubb3*, *Pax2* and *Gad2*), genes that were significantly enriched in W3 (*Foxo1*) and W4 (*Crlf1*), and a proliferation marker (*Mki67*). (I) UMAP of WM NEPs with cellular trajectories embedded (top) or coloured by pseudotime (bottom), computed using scVelo. (J) Heatmap of the normalised activity scores of the top 3 cluster-specific regulons identified by pySCENIC (Table S2). (K) UMAP of WM NEPs coloured by regulon activity. (L,M) Immunofluorescent analysis of P1 *Nes-Cfp*/+ cerebella for CFP, ASCL1, and SOX9 (L) or FOXO1 (M). Arrowheads in L indicate CFP⁺/ASCL1⁻/SOX9⁺ cells; arrowheads in M indicate CFP⁺/ASCL1⁺/FOXO1⁺ triple-positive cells. (N,O) Immunofluorescent analysis of the cerebella of P3 *Ascl1-TdT* pups that were given tamoxifen at P1. Arrowheads indicate TdTOM⁺/SOX9⁺ (N) or TdTOM⁺/FOXO1⁺ (O) cells. Scale bars: 50 µm. OPC/oligo, oligodendrocyte progenitor cells/oligodendrocytes; INs, inhibitory neurons; CN, cerebellar nuclei; GCP, granule cell progenitor; EGL, external granule layer; IGL, internal granule layer; ML, molecular layer; BgL, Bergmann glia layer; WM, white matter.

To this end, we further subclustered specifically the 1234 gliogenic and neurogenic NEPs in the WM (clusters N2, N5, N6 and N8-10) (Fig. 1F,G, Fig. S1I,J and Table S1). We excluded the *Gdf10⁺ Hopx*-NEPs (clusters N0, N1, N4 and N7), as they reside in the BgL and represent a distinct lineage from the WM NEPs (Bayin et al., 2021; Cerrato et al., 2018). This analysis revealed seven distinct clusters, highlighting WM NEP subtypes, one of which was a potential bipotent NEP state (cluster W4, *Ascl1⁺/Hopx*ˡᵒʷ/*Crlf1⁺*) consistent with the previous fate-mapping and clonal analyses (Bayin et al., 2021; Cerrato et al., 2018) (Fig. 1G,H, Fig. S1G). Only one cluster of astrocytes was detected (W6, *Slc6a*11⁺), likely due to their later production compared to neurons. In contrast, we observed five clusters (W0-3 and W5, *Ascl1⁺* and/or *Tubb3⁺*) with neural identity. These clusters differed by expression of transcription factors such as *Foxo1* (W3) and/or their proliferation (W2, W3, W5 and *mKi67⁺*) and maturation (immature inhibitory neurons, W0-1, W5 and *Pax2⁺*, *Gad1/2⁺*) state (Fig. 1G,H, Fig. S1I,J and Table S1). In summary, iterative subclustering of scRNA-seq data resolved distinct cellular states within known NEP subtypes during early postnatal cerebellum development.

### *In silico* analyses predict distinct gene regulatory networks drive NEP lineage progression

To identify the transcriptional mechanisms that drive the differentiation of WM NEPs into inhibitory neurons or astrocytes, we first performed RNA velocity analysis using scVelo (Bergen et al., 2020). As expected, the unsupervised analysis positioned the bipotent cells at the top of the WM NEP lineage hierarchy, leading to two trajectories, either towards inhibitory neurons (*Gad2⁺*) via a *Foxo1⁺* state, or towards astrocytes (*Slc6a11⁺*) (Fig. 1I). Interestingly, various intermediate states were detected along the inhibitory neuron differentiation path, including a subset with persistent *Ascl1* and *Mki67* expression (*Ascl1*-NEPs) (Fig. 1H,I). *Ascl1* expression preceded that of *Foxo1*, albeit the *Ascl1* expression was observed broadly and its levels fluctuated (Fig. S1K). To further explore the molecular drivers of glial and neuronal differentiation, we performed *in silico* GRN reconstruction using pySCENIC (Van de Sande et al.,

2020). This analysis computes an activity score for regulons based on the coordinated expression of a transcription factor and its known target genes. Analysis of the top enriched regulons for each cluster revealed both lineage- and stage-specific transcription factors that could be involved in maintaining stemness, bipotency, lineage decisions or differentiation (Fig. 1J,K and Table S2). For example, the SOX8 regulon was specifically active in bipotent NEPs (cluster W4), whereas LHX1/5 and SOX9 regulons were highly active in immature inhibitory neurons (clusters W1 and W5) and astrocytes (cluster W6), respectively (Fig. 1J,K and Table S2). These findings are in line with previous reports from other brain regions (Kang et al., 2012; Pillai et al., 2007; Seto et al., 2014; Stolt et al., 2003; Sun et al., 2017; Vong et al., 2015). Finally, we hypothesised that the regulons enriched in cluster W3 could play crucial roles during neural differentiation. Among these, FOXO1 was identified as one of the specific regulons in cluster W3 (Fig. 1J,K), overlapping with its high expression in that cluster (Fig. 1H).

We then performed *in silico* gene overexpression and knockout using CellOracle (Kamimoto et al., 2023) (Fig. S1L-Q). Knockout of *Sox8* did not significantly alter the computed cellular trajectories, whereas overexpression of *Sox8* was predicted to promote stemness over differentiation in the bipotent NEPs (Fig. S1M). On the other hand, *in silico* knockout of *Sox9* promoted, whereas knockout of *Ascl1* and *Lhx1* inhibited, neuron differentiation (Fig. S1N-P). In contrast, the overexpression of these transcription factors had opposite effects (Fig. S1N-P). This is in line with their known roles during neural and glial differentiation (Pillai et al., 2007; Stolt et al., 2003; Sudarov et al., 2011; Vong et al., 2015). *In silico* knockout and overexpression of *Foxo1* inhibited and promoted neuronal differentiation, respectively (Fig. S1Q), supporting its potential role during postnatal cerebellar neurogenesis.

To validate the expression of transcription factors identified by scRNA-seq, we analysed sections obtained from P1 *Nes-Cfp*/+ cerebella. Immunofluorescent staining for SOX9, which was enriched in NEPs (W4) and astrocytes (W6), showed expression in a subset of CFP⁺ cells in the WM. Importantly, the expression was mutually exclusive from that of ASCL1 (Fig. 1L). In contrast, we observed nuclear FOXO1 in the CFP⁺ cells in the WM, some of which were also ASCL1⁺, and in endothelial cells, as previously reported (Hosaka et al., 2004) (Fig. 1M). Similarly, fate mapping using *Ascl1*ᶜʳᵉᴱᴿᵀ²/⁺; *R26*ˡᵒˣ-ˢᵀᴼᴾ-ˡᵒˣ-ᵀᵈᵀᵒᵐ/⁺ (*Ascl1-TdT*) animals, where tamoxifen was administered at P1 and analysed 2 days after, revealed only rare SOX9⁺/TdTOM⁺ cells, while 19.94%±0.11% (*n*=3) of TdTOM⁺ cells in the WM had nuclear FOXO1 expression (Fig. 1N,O). FOXO1⁺/TdTOM⁺ cells were restricted to the WM and were not observed in the TdTOM⁺ cells that had migrated to outer layers, highlighting the transient nature of *Foxo1* expression during differentiation. Interestingly, although *Foxo1* mRNA was present in the gliogenic-NEPs in the BgL, immunofluorescent staining on sections from P1 and P5 *Nes-Cfp*/+ cerebella showed cytoplasmic localisation, suggesting limited transcriptional activity in this population (Fig. S1R-U). In summary, using scRNA-seq, we identified GRNs that regulate WM NEP lineage decisions towards astrocytes or neurons, and underscore *Foxo1* as a potential regulator of postnatal cerebellar inhibitory neuron production.

### ASCL1 precedes FOXO1 during postnatal NEP-to-inhibitory neuron differentiation

To investigate the molecular drivers of postnatal NEP-to-inhibitory neuron differentiation, we adopted an *in vitro* approach. Primary NEPs were isolated from P1 cerebella and cultured as neurospheres in suspension with 20 ng/ml EGF and FGF2, similar to other neural

stem cell cultures (Capela and Temple, 2002; Rietze et al., 2001). NEPs could then be dissociated and differentiated as adherent cultures into neurons and glia by withdrawing growth factors and treating with 2% fetal bovine serum (FBS) (Fig. 2A). Almost all cells in the neurospheres were SOX2$^+$ and expressed

either ASCL1 or HOPX, demonstrating the presence of both *Ascl1*- and *Hopx*-NEPs (Fig. 2B). To further validate the composition of our *in vitro* primary NEP cultures and to assess the transcriptional similarities to their *in vivo* counterparts, we performed scRNA-seq of two independent primary NEP

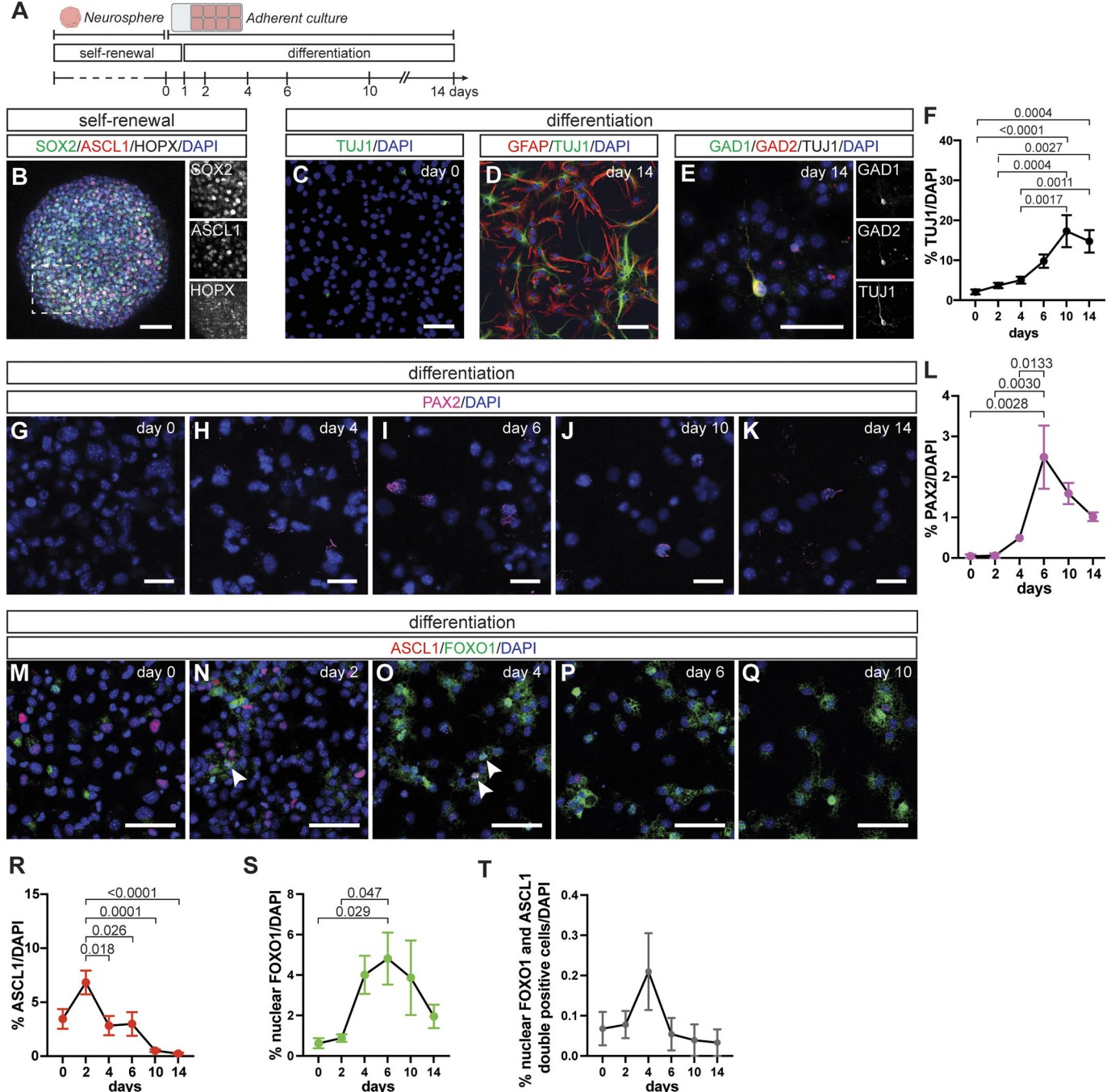

**Fig. 2. NEP-to-inhibitory neuron differentiation and the temporal dynamics of ASCL1 and FOXO1 during differentiation are recapitulated *in vitro*.** (A) Experimental design. (B) Immunofluorescent analysis of primary NEP neurospheres for SOX2 (pan-NEP), ASCL1 (neurogenic-NEP) and HOPX (gliogenic-NEP). (C-E) Immunofluorescent analysis of adherent NEP cultures at day 0 (C) and 14 (D,E) of differentiation. (F) Quantification of TUJ1$^+$ cells during *in vitro* differentiation [one-way ANOVA, $F_{(5,46)}$=9.439, $P$<0.0001, $n$=9 (except day 10, $n$=7)]. (G-K) Immunofluorescent analysis for PAX2 during differentiation. Representative images from days 0 (G), 4 (H), 6 (I), 10 (J) and 14 (K) are shown. (L) Quantification of PAX2$^+$ cells during differentiation [one-way ANOVA, $F_{(5,12)}$=7.973, $P$=0.0016, $n$=3]. (M-Q) Immunofluorescent analysis for ASCL1 and FOXO1 during differentiation. Representative images from days 0 (M), 2 (N), 4 (O), 6 (P) and 10 (Q) are shown. (R-T) Quantification of ASCL1$^+$ (R), nuclear FOXO1$^+$ (S) and nuclear FOXO1 and ASCL1 double-positive (T) cells during differentiation. Arrowheads in N and O highlight nuclear FOXO1 and ASCL1 double-positive cells [one-way ANOVA; R: $F_{(5,46)}$=7.720, $P$<0.0001, $n$=9 (except day 10, $n$=7); S: $F_{(5,46)}$=3.575, $P$=0.0082, $n$=9 (except day 10, $n$=7); T: $F_{(5,46)}$=1.5, $P$=0.1918, $n$=9 (except day 10, $n$=7)]. Data are mean±s.e.m. Scale bars: 50 μm in B-D,M-Q; 25 μm in E,G-K.

neurosphere cultures (total 2606 cells). The scRNA-seq confirmed endogenous *Nes* and *Sox2* expression, and NEP subtype markers *Ascl1* and *Hopx* (Fig. S2A-D, Table S3). Mapping the primary NEP neurosphere scRNA-seq data to the *in vivo* scRNA-seq data of all Nes-CFP$^+$ cells (Fig. 1E) confirmed that the majority of the cells in the neurospheres are transcriptionally similar to WM NEPs/astrocytes (cluster 4) and *Hopx*-NEPs (clusters 0 and 6) (Fig. S2E-G). To further resolve transcriptional similarities within the NEP subtypes, we mapped the *in vitro* datasets to the subsequent iteratively subclustered groups, all NEPs (Fig. 1F, Fig. S2H-J), and the gliogenic and neurogenic NEPs in the WM (Fig. 1G, Fig. S2K-M). These results further confirmed that the primary NEP neurosphere transcriptomes highly resemble WM NEPs and astroglia of the neonatal cerebellum.

Upon *in vitro* differentiation of the primary NEP neurospheres, the number of TUJ1$^+$ neurons gradually increased over time and plateaued, starting at day 10 (Fig. 2C-F). At the end of the 14-day differentiation period, we observed mixed cultures with GFAP$^+$ and TUJ1$^+$ cells (14.74±2.81%, *n*=9) (Fig. 2D-F). Some of the TUJ1$^+$ cells were also GAD1/2$^+$, confirming their inhibitory neuron identity (Fig. 2E). Importantly, when we assessed the expression of PAX2, an immature cerebellar inhibitory neuron marker, during differentiation, we observed a significant increase starting by day 4 and peaking at day 6 of differentiation (Fig. 2G-L). In summary, these results suggest that primary NEP neurospheres maintain their *in vivo* NEP identity and give rise to inhibitory neurons via a PAX2$^+$ intermediate state *in vitro*, providing a model to study cerebellar inhibitory neurogenesis.

We analysed the temporal dynamics of ASCL1 and FOXO1 expression during *in vitro* differentiation (Fig. 2M-S). At day 0 of differentiation, a small proportion of cells were ASCL1$^+$ (3.46%±0.91%, *n*=9), likely representing the *Ascl1*-expressing neurogenic NEPs, whereas nuclear FOXO1$^+$ cells were minimal (0.62%±0.25%, *n*=9) (Fig. 2M,R,S). During differentiation, we first observed a 1.98±0.32-fold (*n*=9) increase in the proportion of ASCL1$^+$ cells on day 2 compared to day 0, which gradually decreased to 0.25%±0.12% (*n*=9) at 14 days (Fig. 2R). On the other hand, the proportion of cells with nuclear FOXO1 increased after the peak of ASCL1$^+$ cells (day 2), starting at around day 4 of differentiation (7.81±1.84-fold compared to day 0, *n*=9) and maintaining similar levels until day 10, with a downwards trend at the end of differentiation at day 14 (Fig. 2S). Although a small population, the proportion of nuclear FOXO1 and ASCL1 double-positive cells also peaked at day 4 of differentiation, aligning with the onset of the FOXO1$^+$ state (Fig. 2T). Interestingly, the broad *Ascl1* mRNA expression observed via scRNA-seq (Fig. S1K) was not mimicked by the ASCL1 protein dynamics (Fig. 2R), highlighting potential non-transcriptional mechanisms to control ASCL1 levels. Collectively, these results demonstrate that ASCL1 precedes FOXO1 during *in vitro* differentiation of NEPs into inhibitory neurons, in agreement with the scRNA-seq and *Ascl1-TdT* fate-mapping results.

### ASCL1 and FOXO1 both drive neurogenesis *in vitro*

*In vivo* studies have shown that ASCL1 is required for inhibitory neuron production in the developing cerebellum (Sudarov et al., 2011). However, the role of FOXO1 during NEP-to-inhibitory neuron differentiation, and whether ASCL1 and FOXO1 interact during differentiation remain to be understood. To resolve their functions, we performed gain- and loss-of-function studies in our *in vitro* culture paradigm (Fig. 3A,L). To test whether ASCL1 and FOXO1 alone can promote neurogenesis, we used a doxycycline (DOX)-inducible lentiviral system to overexpress them in a

temporally restricted manner (Fig. 3A). We established a stable rTTA-NEP cell line, which was then infected with lentiviruses that overexpressed either ASCL1 (ASCL1 OE), FOXO1 (FOXO1 OE) or green fluorescent protein (EGFP OE) under the control of a TET-responsive promoter (Liu et al., 2018; Panciera et al., 2016) (Fig. 3A). Quantification of EGFP$^+$ cells in the EGFP OE NEPs demonstrated transduction efficiencies of 30.33±2.38% (*n*=5) at day 4 and 33.88±1.61% (*n*=5) at day 10 of differentiation (Fig. 3B,C, Fig. S3A,E). Unfortunately, the TET-inducible promoter was leaky in primary NEPs, and we detected similar frequencies of EGFP$^+$ cells with or without DOX (Fig. S3A,E). Therefore, we focused on DOX-treated cells for downstream analyses. No significant changes were observed in the parameters assessed between cells treated with DOX or vehicle (Fig. S3A-H). However, the level of overexpression was higher with DOX (Fig. S3I-K).

We confirmed overexpression by quantifying the percentage of ASCL1$^+$ and nuclear FOXO1$^+$ cells in each overexpression condition during differentiation. On day 4, all conditions showed significant upregulation of the respective protein (Fig. 3B-G). Interestingly, ASCL1 OE also showed a significant increase in nuclear FOXO1$^+$ cells (Fig. 3G), suggesting that ASCL1 promotes FOXO1 expression. Next, we addressed the effect of overexpression on neurogenesis by quantifying the number of neurons (TUJ1$^+$) in each condition during differentiation (Fig. 3H-K). We observed a significant increase in the proportion of neurons upon both ASCL1 OE and FOXO1 OE at day 4, with the fold-change relative to EGFP OE being higher in ASCL1 OE compared to FOXO1 OE (2.46±0.16-fold versus 1.54±0.18-fold, *n*=5, Fig. 3J). Surprisingly, while the FOXO1 OE condition continued to have significantly more TUJ1$^+$ cells at day 10 of differentiation, the ASCL1 OE did not (Fig. 3K). Furthermore, the TUJ1$^+$ cells in the ASCL1 OE condition had longer processes compared to the EGFP OE and FOXO1 OE conditions both at day 4 and 10, suggesting that sustained overexpression of these factors that function at the early stages of differentiation may affect neural maturation or even lead to cell death.

To test whether FOXO1 is required for NEP-to-inhibitory neuron differentiation, we performed lentiviral knockdown using shRNAs against *Foxo1* during *in vitro* differentiation. Stable primary NEP lines expressing either shRNAs against *Foxo1* or a scrambled shRNA as a control were generated and then differentiated (Fig. 3L). Quantification of FOXO1$^+$ cell numbers showed a significant but partial reduction in the percentage of FOXO1$^+$ cells (FOXO1 shRNA1, 0.41±0.07-fold; FOXO1 shRNA2, 0.47±0.07-fold; *n*=6), confirming the knockdown (Fig. 4M,N). FOXO1 knockdown led to a significant reduction in the percentage of TUJ1$^+$ cells at day 10 of differentiation (shRNA1, 0.73±0.08-fold; shRNA2, 0.72±0.09-fold; *n*=6), while the number of ASCL1$^+$ cells did not change at day 4 (Fig. 3O-Q). Together, our *in vitro* functional analyses demonstrate that both ASCL1 and FOXO1 promote the production of neurons *in vitro* but to different extents and suggest that ASCL1 regulates FOXO1 expression during the differentiation of NEPs to inhibitory neurons.

### ASCL1 regulates FOXO1 expression, and ASCL1 and FOXO1 have shared and distinct binding targets during *in vitro* differentiation

Having established that ASCL1 and FOXO1 are important for NEP-to-inhibitory neuron differentiation, we took a targeted DNA adenine methyltransferase identification (Targeted DamID; TaDa) approach (Marshall and Brand, 2015; Marshall et al., 2016; Southall et al., 2013) to identify their direct regulatory targets. Primary NEPs

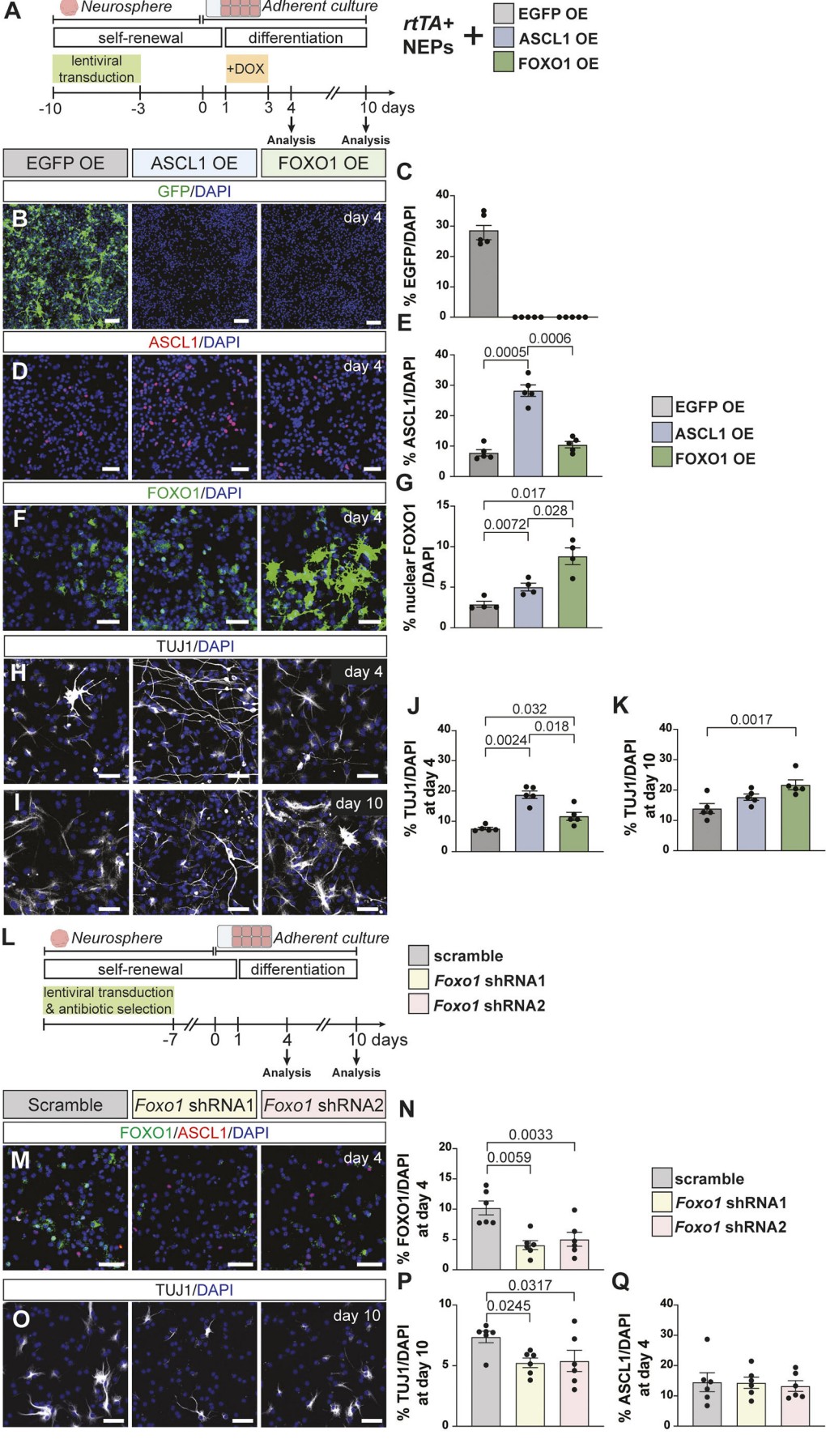

**Fig. 3. FOXO1 is downstream of ASCL1 and promotes neuron production *in vitro*.**
(A) Experimental design.
(B-G) Immunofluorescent analysis of differentiating cells at day 4 that are overexpressing EGFP (EGFP OE), ASCL1 (ASCL1 OE) or FOXO1 (FOXO1 OE). EGFP⁺ (B,C), ASCL1⁺ (D,E) or FOXO1⁺ (F,G) cells were quantified [one-way ANOVA; C: $F_{(1,4)}$=149, $P$=0.0003, $n$=5; E: $F_{(1,5)}$=138.9, $P$<0.0001, $n$=5; G: $F_{(1,3)}$=35.83, $P$=0.0087, $n$=4].
(H,I) Immunofluorescent analysis of EGFP OE, ASCL1 OE and FOXO1 OE cultures at day 4 (H) or 10 (I) of differentiation. (J,K) TUJ1⁺ cells were quantified on day 4 (J) and 10 (K) [one-way ANOVA; J: $F_{(1,6)}$=39.25, $P$=0.0002, $n$=5; K: $F_{(1,5)}$=17.37, $P$=0.0044, $n$=5].
(L) Experimental design.
(M) Immunofluorescent analysis of cultures overexpressing scramble, *Foxo1* shRNA1 or *Foxo1* shRNA2 for FOXO1 and ASCL1 at day 4 of differentiation. (N) Quantification of FOXO1⁺ cells at day 4 in the scramble, *Foxo1* shRNA1- or *Foxo1* shRNA2-expressing cells ($n$=6, paired Student's *t*-test, compared to scrambled). (O) Immunofluorescent analysis of cells that are expressing scramble, *Foxo1* shRNA1 or *Foxo1* shRNA2 for TUJ1 at day 10. (P,Q) Quantification of TUJ1⁺ cells at day 10 (P) or ASCL1⁺ cells at day 4 (Q) ($n$=6, paired Student's *t*-test, compared to scrambled). Representative images are shown. Data are mean±s.e.m. Scale bars: 50 µm.

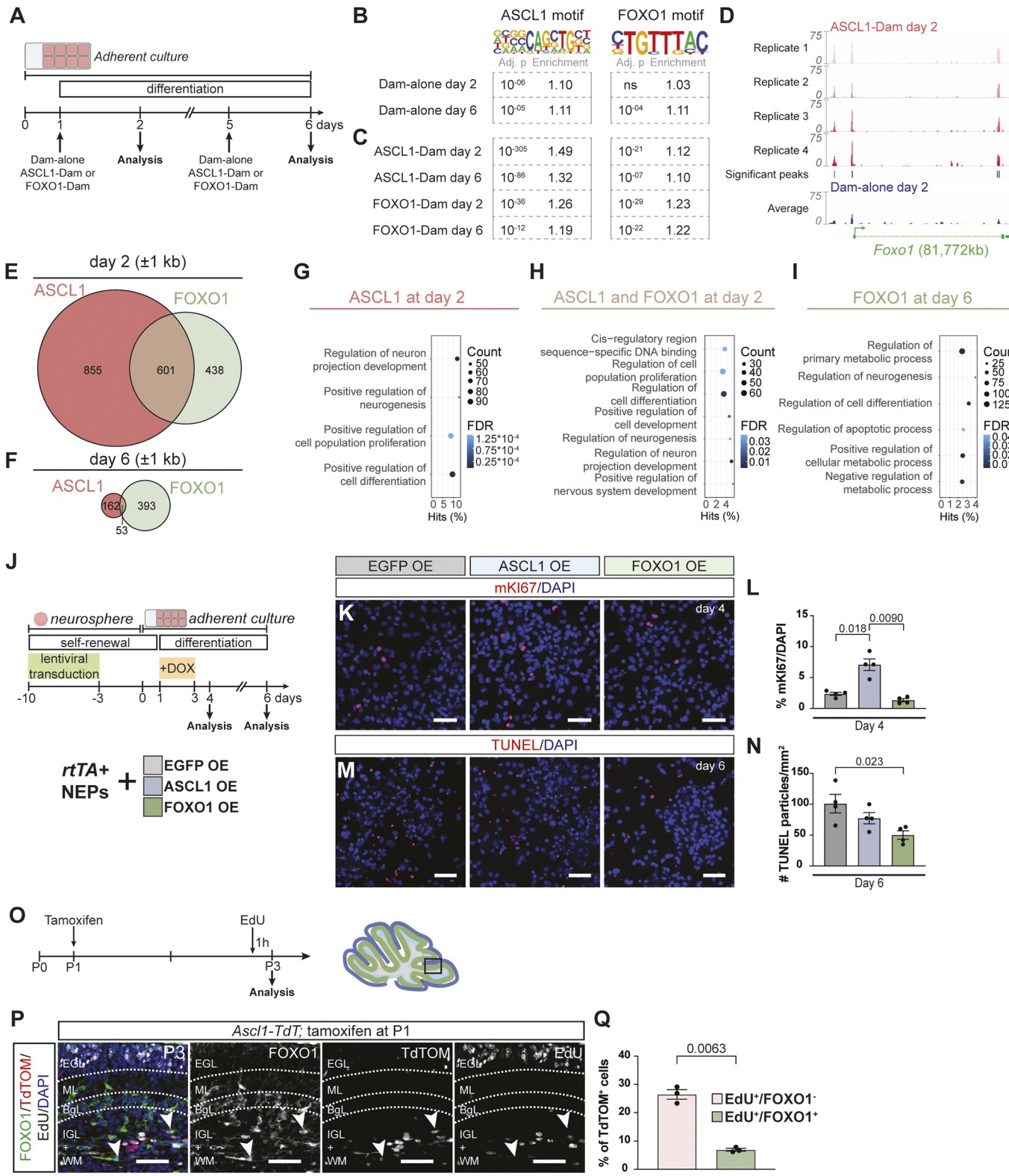

**Fig. 4.** See next page for legend.

were infected with lentiviral constructs for expression of untethered Dam (Dam-alone) or fusions of either ASCL1 or FOXO1 with Dam (ASCL1-Dam and FOXO1-Dam) 1 day before analysis on day 2 and 6 of differentiation (Fig. 4A). Time points were decided based on peak ASCL1 and FOXO1 expression during differentiation (Fig. 2R,S). After normalisation of fusion protein data to their respective Dam-alone control, significant peaks were calculated. 11,267 and 2537 significant ASCL1-Dam peaks were identified at day 2 and day 6, respectively, with an average width of ∼431 and ∼454 bp. Meanwhile, 9717 and 4211 significant FOXO1-Dam peaks were identified at day 2 and day 6, respectively, with an average width of ∼362 and ∼488 bp (Fig. S4A). When the

**Fig. 4. ASCL1 and FOXO1 independently regulate proliferation and cell survival during *in vitro* NEP differentiation.** (A) Experimental design. (B,C) Table of HOMER results showing ASCL1 and FOXO1 binding motifs, and their significance (*P*_adj) and enrichment score in respective conditions at day 2 and 6 (Table S4). Analysis was performed on significant Dam-alone peaks as a readout for open chromatin (B) and on significant ASCL1-Dam and FOXO1-Dam peaks, which previously had been normalised to their Dam-alone control (C). (D) ASCL1-Dam tracks and the average of Dam-alone track (*n*=4) on day 2 of differentiation at the *Foxo1* loci. Significant ASCL1-Dam peaks are shown (FDR<0.05). (E,F) Venn diagrams of genes associated with significant and reproducible ASCL1-Dam and FOXO1-Dam peaks ±1 kb of their transcriptional start site on day 2 (E) and 6 (F) of differentiation (Table S5). (G-I) Selected significantly enriched GO terms for genes with ASCL1-Dam binding at day 2 (G), ASCL1-Dam and FOXO1-Dam binding at day 2 (H) or FOXO1-Dam binding at day 6 (I) are shown (Table S5). The circle size indicates the number of genes from a given GO term that contains a significant peak. The *x*-axis is the percentage of all the genes within a GO term with a significant peak. The colour gradient indicates the false discovery rate. (J) Experimental design. (K,L) Immunofluorescent analysis and quantification of EGFP OE, ASCL1 OE and FOXO1 OE cultures at day 4 for proliferation marker mKi67 [one-way ANOVA, $F_{(1, 3)}$=46.00, *P*=0.0029, *n*=4]. (M,N) TUNEL staining and quantification of TUNEL$^+$ particles on EGFP OE, ASCL1-OE and FOXO1-OE cultures at day 6 [one-way ANOVA, $F_{(2, 9)}$=5.475, *P*=0.0278, *n*=4]. (O) Experimental design. (P) Immunofluorescent analysis of P3 *Ascl1-TdT* brains that were given tamoxifen at P1 and EdU 1 h before the experimental endpoint. Arrowheads indicate TdTOM$^+$/FOXO1$^+$/EdU$^-$ cells. (Q) Quantification of EdU$^+$ and EdU$^+$/nuclear FOXO1$^+$ cells in all TdTOM$^+$ cells at P3 (*n*=3, paired Student's *t*-test). Representative images are shown. Data are mean±s.e.m. Scale bars: 50 μm.

chromosomal distribution of the peaks was analysed, the majority resided in potential regulatory regions (promoters, introns and intergenic) and the distribution did not vary across conditions and time (Fig. S4B,C).

The peaks observed in Dam-alone condition reflect open chromatin (Aughey et al., 2018). Therefore, we first tested whether ASCL1 and FOXO1 motifs are enriched in open chromatin. Motif analysis of Dam-alone peaks showed that ASCL1 motif was enriched in Dam-alone on both day 2 and 6 of differentiation (day 2, *P*_adj<10$^{-6}$; day 6, *P*_adj<10$^{-5}$, Fig. 4B, Table S4). On the other hand, the FOXO1 motif was only significantly enriched on day 6 Dam-alone peaks (*P*_adj<10$^{-4}$, Fig. 4B), potentially highlighting a later role during differentiation. These results show that some ASCL1 and/or FOXO1 target genes are accessible under differentiation conditions *in vitro*, suggesting involvement of upstream chromatin modifiers as NEPs commit to differentiation.

To validate faithful binding of ASCL1-Dam and FOXO1-Dam, and to test which other transcription factor motifs are detected in individual conditions, we next performed motif enrichment analysis on ASCL1-Dam and FOXO1-Dam peaks on both days following normalisation to Dam-alone peaks (Fig. 4B,C, Table S4). While normalisation to Dam alone allows the identification of most significantly bound loci, this could conceal the extent of enrichment. Nevertheless, analysis of motifs in ASCL1-Dam and FOXO1-Dam peaks revealed a significant enrichment of the ASCL1 and FOXO1 motifs within the respective significant peaks at both timepoints. Importantly, when these were compared to Dam alone, greater significance (lower *P*_adj values) and higher enrichment scores were observed (compare Fig. 4C with 4B, and Table S4). In addition, both transcription factors were significantly enriched within each other's peaks, as previously reported (Fig. 4C) (Park et al., 2017). Finally, when we analysed other significant motifs in ASCL1-Dam and FOXO1-Dam peaks, we observed co-occurrence of other bHLH neurogenic transcription factors, SOX family genes, TCF4 and AP-1 consistently (*P*_adj<0.05, Table S4). Together, these

results show that the chromatin state is dynamic during NEP *in vitro* differentiation. Additionally, the ASCL1 and FOXO1 motifs are significantly enriched in open chromatin, concurrent with their expression patterns, while the fusion proteins further enrich for motif-containing sequences, confirming their faithful binding.

ASCL1 OE NEPs showed a significant upregulation of FOXO1 during *in vitro* differentiation (Fig. 3G), suggesting that FOXO1 may be a direct transcriptional target of ASCL1. Therefore, we assessed the *Foxo1* locus for ASCL1-Dam binding. On day 2, ASCL1 binding was present at the FOXO1 promoter, indicating direct regulation of FOXO1 expression (Fig. 4D). Partial knockdown of ASCL1 *in vitro* via shRNAs during NEP-to-inhibitory neuron differentiation and the analysis of CKO mice, where *Ascl1* is removed in *Ascl1*-NEPs postnatally using *Ascl1$^{CreERT2/fl}$* animals, showed a significant reduction in the number of FOXO1$^+$ cells both *in vitro* and *in vivo*, compared to scramble shRNA or *Ascl1$^{fl/fl}$* littermate controls, respectively (Fig. S5A-H). These results further confirm that FOXO1 expression is regulated by ASCL1 in neurogenic NEPs during postnatal cerebellum development.

To resolve the targets of ASCL1 and FOXO1, we next identified all genes that showed significant ASCL1- or FOXO1-binding peaks within ±1 kb of their transcriptional start site (TSS) (Fig. 4E,F). We observed that more genes were bound by ASCL1 and/or FOXO1 on day 2 compared to day 6 (ASCL1, 1456 versus 215 genes; FOXO1, 1039 versus 446) (Fig. 4E,F). Interestingly, on day 2 of differentiation, a greater proportion of genes were associated with significant ASCL1 binding (1.40-fold compared to FOXO1), whereas this was inverted on day 6, with FOXO1 being associated with more genes (2.07-fold compared to ASCL1) (Fig. 4E,F). This suggests that ASCL1 has a more dominant role at early stages of differentiation, whereas FOXO1 has a more dominant role at later stages, in line with their temporal expression patterns that we identified *in vitro* (Fig. 2R,S). Importantly, we observed a 31.73% and 8.72% overlap between genes [29.81% and 8.41% (peaks ±1 kb of a TSS), 31.20% and 8.69% (peaks ±2 kb of a TSS) and 37.85% and 13.57% (all peaks)] bound by both ASCL1-Dam and FOXO1-Dam ±1 kb of their TSS on days 2 and 6 of differentiation, respectively (Fig. 4E,F). When motif enrichment analysis was performed on Dam-alone peaks associated with the genes that showed both ASCL1-Dam and FOXO1-Dam binding, we observed significant enrichment for only a few motifs at day 2 (<15 motifs with *P*_adj<0.05), which include some bHLH neurogenic transcription factors (Table S4). No significant motifs were detected at day 6 (Table S4), likely due to the low number of genes used for this analysis (*n*=53 genes for peaks with ±1 kb of TSS and *n*=90 genes for peaks with ±2 kb of TSS). Collectively, these results highlight a potential cooperation between ASCL1 and FOXO1 during differentiation, though whether this is synergistic or antagonistic, and what additional factors may be involved, remains to be determined.

To provide further insights into the functional roles of genes bound by ASCL1 and FOXO1, we performed gene ontology analysis using PANTHER (Thomas et al., 2022) (Fig. 4G-I). Due to the low number of genes bound by ASCL1 and both ASCL1/FOXO1 on day 6, there were no significantly enriched GO terms identified (Table S5). Given the known role of ASCL1 during neurogenesis, it was unsurprising that genes bound by ASCL1 on day 2 were linked to GO terms related to regulation of nervous system development, neurogenesis and proliferation (Fig. 4G). These terms were also enriched for genes bound by both ASCL1 and FOXO1 at day 2 (Fig. 4H). In contrast, genes bound by only FOXO1 at day 6 were enriched in GO terms related to metabolic and apoptotic processes (Fig. 4I, Table S5). These findings are in

agreement with previous studies showing that FOXO1 regulates metabolic reprogramming, cell survival and apoptosis in other cell types (Bastie et al., 2005; Brunet et al., 2004; De La Torre-Ubieta et al., 2010; Puigserver et al., 2003; Schäffner et al., 2018; Yuan et al., 2008). We also repeated the same analysis with genes that showed significant ASCL1 or FOXO1 binding peaks within ±2 kb of their TSS and did not observe the emergence of any new GO terms, suggesting additional roles for ASCL1 and FOXO1 (Fig. S4D,E and Table S5). Collectively, these data outline key overlapping and independent roles of ASCL1 and FOXO1 during neurogenesis. During the early stages of differentiation, FOXO1 cooperates with ASCL1 by targeting neurogenic genes, but later diverges in binding to target genes involved in survival and maturation.

Finally, to orthogonally validate the ASCL1-Dam and FOXO1-Dam target genes *in vivo*, we performed *in situ* hybridisation chain reaction (HCR) for 2-4 genes that had a significant peak in either ASCL1-Dam (*Dkk2* and *Dlll3*) or FOXO1-Dam (*Cdkn1b*, *Gpx4* and *Akt1s1*) or both (*Id1*, *Fzd4*, *Axin2* and *En1*) on cerebellum sections obtained from P3 *Ascl1-TdT* pups that were given tamoxifen at P1 to allow identification of *Ascl1*-NEP progeny (Fig. S6A-I). As expected, target gene expression within the TdTOM+ cells overlapped only with their respective transcription factors, such that the expression of ASCL1-Dam target genes overlapped with only *Ascl1* expression (Fig. S6A,B), FOXO1-Dam target genes with only *Foxo1* expression (Fig. S6G-I), and genes targeted by both ASCL1-Dam and FOXO1-Dam overlapped with both *Ascl1* and *Foxo1* expression (Fig. S6C-F).

## ASCL1 and FOXO1 regulate distinct cellular processes during differentiation

Our analysis suggests that ASCL1 and FOXO1 cooperate during NEP-to-inhibitory neuron differentiation, regulating different cellular processes to facilitate inhibitory neuron production. Based on the GO term analysis of ASCL1 and FOXO1 bound genes, we hypothesised that, alongside their roles in neurogenesis, ASCL1 may regulate proliferation and FOXO1 may regulate cell survival. To test this hypothesis, we assessed proliferation and apoptosis in ASCL1 OE and FOXO1 OE conditions compared to the EGFP OE control (Fig. 4J). While ASCL1 OE showed a significant increase in the percentage of mKI67+ cells at day 4 of differentiation (2.99±0.40-fold compared to EGFP OE, *n*=4), FOXO1 OE did not affect proliferation (Fig. 4K,L). On the other hand, FOXO1 OE cells showed a significant decrease in TUNEL+ particles compared to the EGFP OE control (0.50±0.07-fold, *n*=4), while ASCL1 OE had no effect (Fig. 4M,N). These data suggest that the differentiation of NEPs to inhibitory neurons occurs via a proliferative neurogenic ASCL1+ state that transitions into a FOXO1+ state, which then promotes neuronal survival and maturation. Indeed, analysis of proliferation in P3 *Ascl1-TdT* cerebella that were given tamoxifen at P1 and EdU 1 h before analysis revealed that, of the TdTOM+ cells, FOXO1+ cells showed a significant 0.26±0.018-fold (*n*=3) reduction in EdU incorporation compared to FOXO1− cells (Fig. 4O-Q). This analysis confirmed the change in the proliferative status of NEPs from an ASCL1+ to a FOXO1+ state during differentiation. Finally, loss of *Ascl1* in the *Ascl1*-NEPs also led to a significant reduction in the density of mKi67+ cells in P3 *Ascl1*CreERT2/fl pups that were given tamoxifen at P0, compared to their *Ascl1*fl/fl littermate controls (Fig. S5I). In summary, these results demonstrate that during NEP-to-inhibitory neuron differentiation, ASCL1 promotes proliferation and FOXO1 supports cell survival, also explaining why ASCL1 OE promoted more neurogenesis than FOXO1 OE *in vitro* (Fig. 3J).

## Activation of WNT signalling promotes a FOXO1+ state and increases neurogenesis in primary NEP cultures

It remained unclear what promotes the transition of *Ascl1*-NEPs from the ASCL1+ state to a FOXO1+ state during differentiation. To explore this, we revisited our scRNA-seq data to identify signalling pathways with activation patterns overlapping with *Foxo1* expression and regulon activity (Fig. 1H,K). To this end, we utilised Cell2Fate (Aivazidis et al., 2025), an algorithm optimised to reconstruct cellular trajectories from scRNA-seq data, which orders cells according to an inferred time and computes modules of genes that follow the same mRNA splicing dynamics (Fig. 5A,B and Fig. S7A). This allowed for better pseudotemporal ordering of cells after the neurogenic commitment of WM NEPs compared to scVelo (clusters W0-3 and 5, Fig. 5A compared to Fig. 1I) and facilitated the interrogation of the commonalities between the genes driving the computed inferred time. Interestingly, Cell2Fate identified the immature astrocytes (cluster W6) as the top of the hierarchy. Whether the immature astrocytes in the neonatal cerebellum retain WM NEP identity and biopotency remains to be investigated.

Cell2Fate generates modules of genes that follow the same mRNA splicing dynamics (Aivazidis et al., 2025). The state of each module is classified for individual cells based on how the amount of spliced UMIs for genes within a module change with respect to the inferred time. 'Induction' or 'repression' designates whether the amount of spliced UMIs increases or decreases with time, respectively. If the total amount of spliced UMIs is <5% or >95% of the maximal steady-state counts, the modules are classified as 'Off' or 'On' (Aivazidis et al., 2025). Nine modules with distinct pseudotemporal dynamics were identified (Fig. 5C-E, Fig. S7B,C and Table S6). Of these modules, we specifically focused on modules 2, 4 and 6. Module 2 aligned with WM NEPs (cluster W4), whereas modules 4 and 6 induction overlapped with the *Foxo1*+ state (cluster W3) or was induced immediately after the *Foxo1*+ state (clusters W2 and W0) (Fig. 5C-E). GO term analysis of the top 200 module markers revealed an upregulation of WNT signalling prior to the *Foxo1*+ state (module 2) (Fig. 5F). This was followed by an enrichment of neurogenesis-related genes (module 4) and then genes involved in negative regulation of WNT signalling and cell migration/motility (module 6) (Fig. 5G,H and Table S6). Interestingly, module 2 also showed enrichment for GO terms related to negative regulation of WNT, albeit with lower enrichment (Table S6), suggesting a potential negative-feedback regulation of the pathway within these cells. In summary, these data suggest that transient activation of the WNT signalling pathway may facilitate the progression of NEP-to-inhibitory neuron differentiation during postnatal development.

To test whether WNT signalling promotes the transition of *Ascl1*-NEPs into the *Foxo1*+ state, we activated canonical WNT signalling during *in vitro* differentiation with CHIR99021 (CHIR) (Fig. 5I). CHIR treatment during differentiation led to a significant 1.55±0.27-fold (*n*=6) increase in the percentage of cells with nuclear FOXO1 at day 4 but did not significantly affect the number of ASCL1+ or mKI67+ cells (Fig. 5J-M and Fig. S7D). Although not significant, we observed a trend towards an increase in the number of nuclear FOXO1 and ASCL1 double-positive cells (Fig. 5N). CHIR treatment also resulted in a significant 1.62±0.17-fold (*n*=6) increase in the percentage of TUJ1+ cells produced after 10 days of differentiation (Fig. 5O-Q). Collectively, these data suggest that WNT signalling promotes the transition from the ASCL1+ state to the FOXO1+ state during *Ascl1*-NEP differentiation to inhibitory neurons during postnatal cerebellar development.

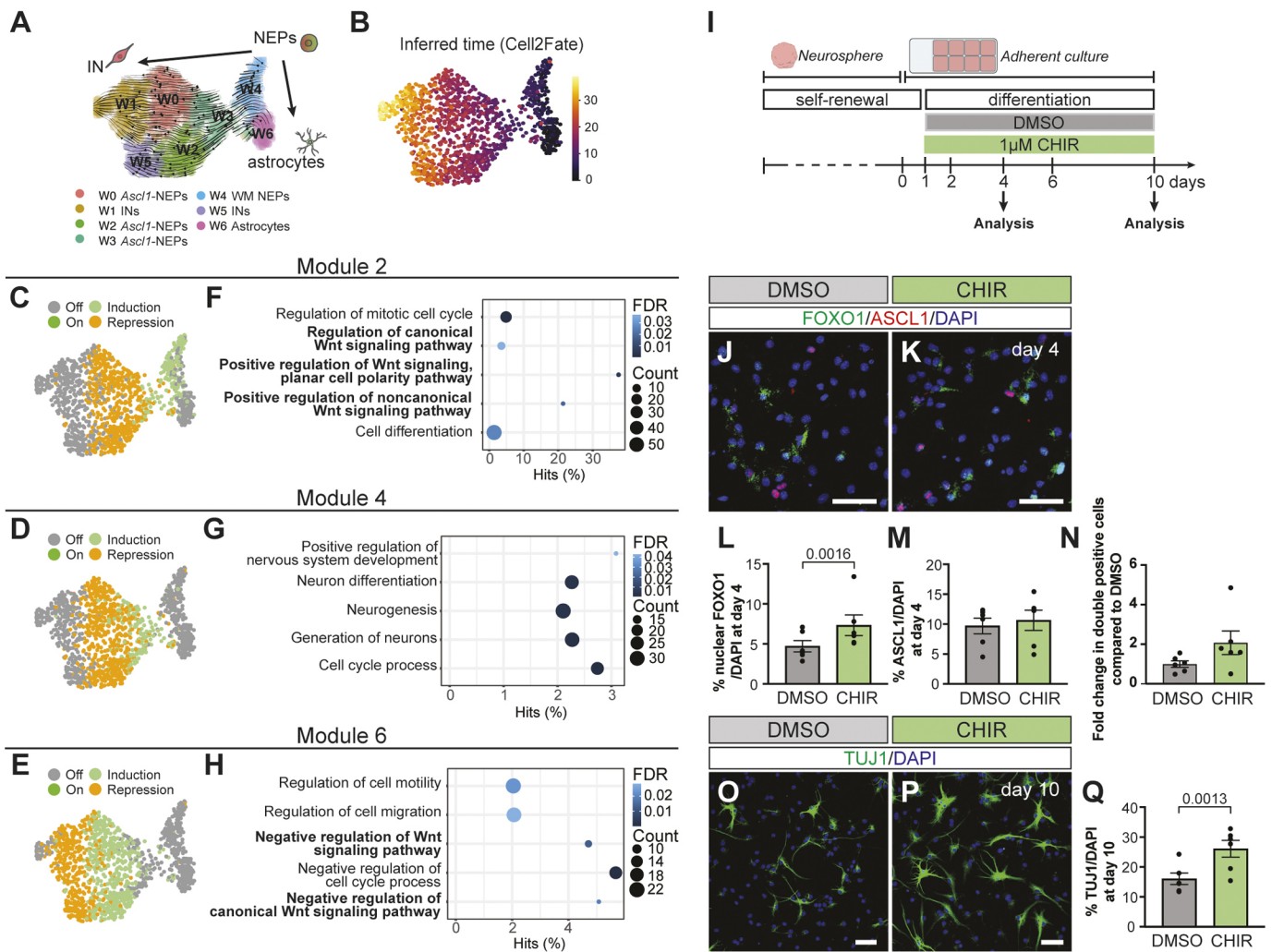

**Fig. 5. WNT signalling is a positive regulator of FOXO1 and neural production during NEP-to-inhibitory neuron differentiation.** (A,B) UMAP projection of cellular trajectories (A) or inferred time (B) computed using Cell2Fate within the gliogenic and neurogenic NEPs in the WM. (C-E) UMAP projections showing the state of modules 2 (C), 4 (D) or 6 (E). (F-H) GO term analysis of module genes from modules 2 (F), 4 (G) or 6 (H). Selected significantly enriched GO terms are shown (Table S6). The circle size indicates the number of genes from a given GO term. The x-axis is the percentage of all the genes within a GO term. The colour gradient indicates the false discovery rate. (I) Experimental design. (J-M) Immunofluorescent analysis (J,K) and quantification (L,M) of nuclear FOXO1⁺ (L) and ASCL1⁺ (M) cells in cultures treated with DMSO or CHIR at day 4 of differentiation (n=6, paired Student's t-test). (N) Fold change in % nuclear FOXO1 and ASCL1 double-positive cells in cultures treated with CHIR compared to DMSO (n=6, paired Student's t-test). (O-Q) Immunofluorescent analysis (O,P) and quantification (Q) of TUJ1⁺ cells in cultures treated with DMSO or CHIR at day 10 of differentiation (n=6, paired Student's t-test). Representative images are shown. Data are mean±s.e.m. Scale bars: 50 µm.

## The ASCL1-FOXO1 axis is conserved during human NEP-to-inhibitory neuron differentiation *in vitro*

To assess whether the molecular mechanisms that promote mouse molecular layer inhibitory neuron differentiation are conserved in the human cerebellum, we utilised human fetal cerebellar tissue to study the expression of ASCL1 and FOXO1. Immunofluorescent analysis on histological sections from a 17 postconception week (pcw, an equivalent stage to around birth in mice; Haldipur et al., 2022) human cerebellum revealed nuclear FOXO1 expression in SOX2⁺ cells in the prospective WM, some of which were also ASCL1⁺. This suggested that FOXO1 is expressed in human *ASCL1*-NEPs (hNEPs) (Fig. 6A).

To investigate the dynamics of ASCL1 and FOXO1 expression during hNEP differentiation, we established primary NEP cultures from human fetal tissue ranging from 9-17 pcw. Similar to primary mouse NEPs, these cells were also maintained as neurospheres and were differentiated as adherent cultures (Fig. 6B). Immunofluorescent

staining of hNEP neurospheres revealed that most cells expressed SOX2, and both HOPX⁺ and ASCL1⁺ hNEPs were present (Fig. 6C). To further characterise the cellular composition of primary hNEP neurosphere cultures and to test the transcriptional similarities to human cerebellar cells, we performed scRNA-seq of 10 different primary hNEP neurosphere cultures (ages ranging from 9-17 pcw). Clustering of 10,143 cells after filtering revealed nine clusters, three of which were of mesenchymal origin (clusters 0-1 and 7, *DCN*⁺, Fig. S8A-D), demonstrating a higher cellular diversity than the mouse primary NEP cultures (Fig. S2A-D and Table S7). In the neuronal cells (clusters 2-5, Fig. S8B), we observed endogenous expression of *NES* and *SOX2*, along with NEP subtype markers *HOPX* and *ASCL1*, likely reflecting the ventricular zone origin of the cells and their NEP identity (Fig. S8D). In order to further confirm the transcriptional similarities between primary hNEP cultures and human cerebellar NEPs, we re-analysed a previously published snRNA-seq data from

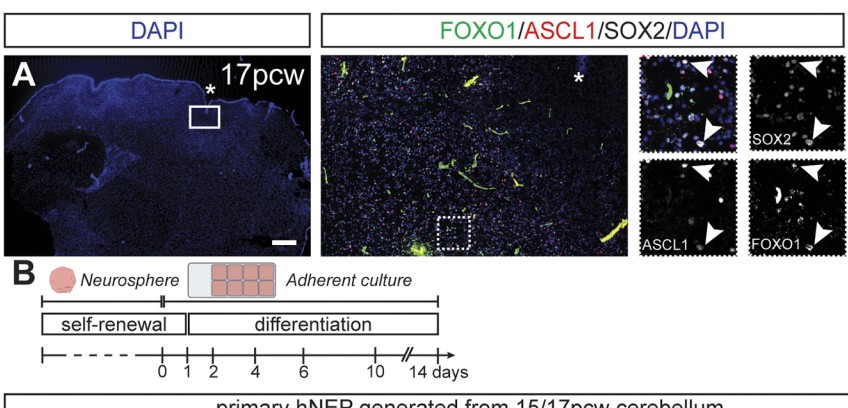

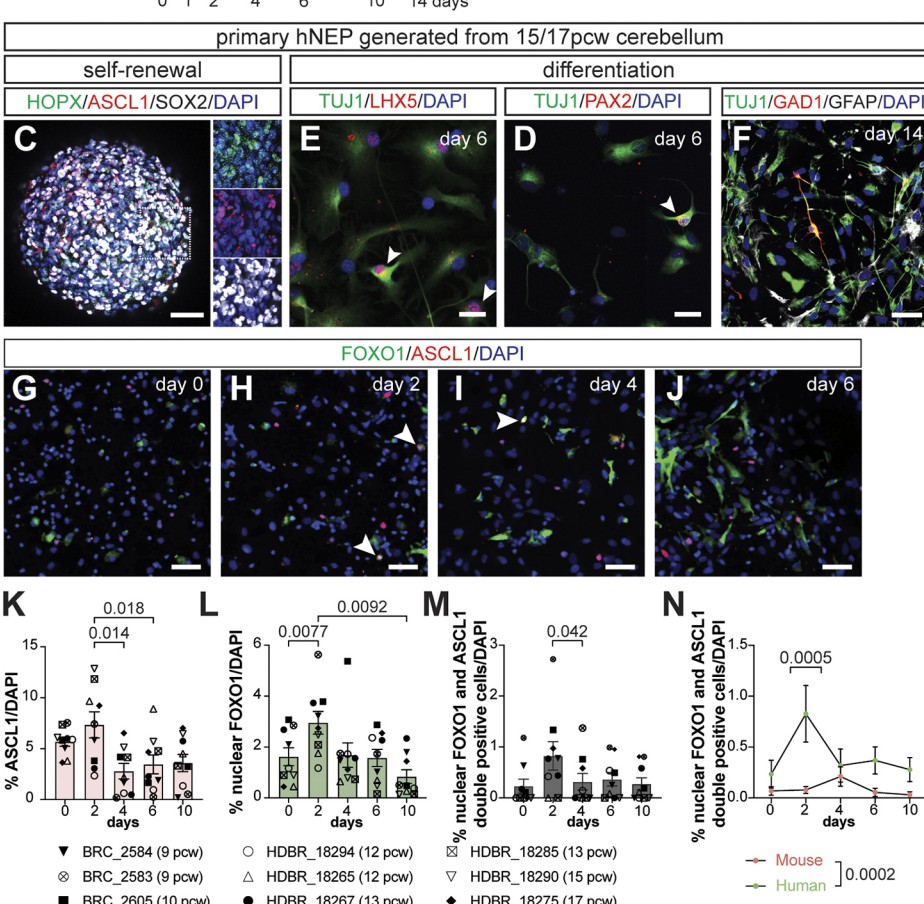

Fig. 6. ASCL1 and FOXO1 are expressed during human NEP-to-inhibitory neuron differentiation. (A) Immunofluorescent analysis of a 17 pcw human cerebellum. Arrowheads show FOXO1⁺/ASCL1⁺/SOX2⁺ triple-positive cells; asterisks indicate an anchoring centre. (B) Experimental design. (C-F) Immunofluorescent analysis of primary hNEP neurospheres showing SOX2⁺, HOPX⁺ and ASCL1⁺ cells (C), 2D cultures during differentiation showing PAX2⁺ (D), and TUJ1⁺ and LHX5⁺ (E) cells at day 6; and GFAP⁺, TUJ1⁺ and GAD1⁺ cells at day 14 (F). Insets in C show higher magnification images. Arrowheads highlight LHX5⁺ or PAX2⁺ cells. (G-J) Immunofluorescent analysis for ASCL1 and FOXO1 during hNEP differentiation. Representative images of day 0 (G), 2 (H), 4 (I) and 6 (J) are shown. Arrowheads highlight nuclear FOXO1 and ASCL1 double-positive cells. (K-M) Quantification of ASCL1⁺ (K), nuclear FOXO1⁺ (L) and nuclear FOXO1 and ASCL1 double-positive (M) cells throughout *in vitro* differentiation [one-way ANOVA; K: $F_{(2, 23)}$=6.614, $P$=0.0023, $n$=9; L: $F_{(2, 20)}$= 6.812, $P$=0.0033, $n$=9; M: $F_{(2, 18)}$=5.104, $P$-value=0.0141]. (N) Quantification of nuclear FOXO1 and ASCL1 double-positive cells during *in vitro* differentiation of mouse and human NEPs [two-way ANOVA, $F_{(1, 80)}$=14.86, $P$=0.0002, $n$=9 (mouse, except day 10 $n$=7) or $n$=9 (human)]. Data are mean±s.e.m. Scale bars: 50 μm in C-J; 500 μm in A.

human fetal cerebellum (Sepp et al., 2024), focusing on 17 and 20 pcw datasets to reflect mouse cerebellar development around birth. Reclustering of the 17 and 20 pcw Sepp et al. dataset demonstrated the expected cerebellar cell types, including *NES*⁺ and *SOX2*⁺ progenitors, and inhibitory (*PAX2*⁺/*GAD1*⁺) and excitatory neurons (*PAX6*⁺/*RBFOX3*⁺) (Fig. S8E-H and Table S7). Mapping of the *in vitro* dataset to the *in vivo* snRNA-seq dataset showed that the majority of the neuronal cells (cluster 2-5) from the primary hNEP neurosphere cultures exhibit high transcriptional similarity to *NES*- and *SOX2*-expressing clusters (Fig. S8I,J). While further analysis is needed to resolve the cellular diversity within hNEP neurosphere cultures across different fetal stages, these results highlight their NEP-like identity. Following *in vitro* differentiation for 14 days, we obtained a mixed culture of GFAP⁺ astrocytes and TUJ1⁺ neurons, some of which were GAD1⁺ inhibitory neurons (Fig. 6E,F). Importantly, analyses of PAX2 and LHX5 during hNEP *in vitro* differentiation further confirmed the cerebellar inhibitory

neuron identity (Fig. 6E-D). Collectively, these data confirm that primary hNEPs resemble their *in vivo* counterparts and can also generate inhibitory neurons *in vitro*, providing us with a reliable model to study differentiation mechanisms.

We assessed the percentage of ASCL1⁺ and nuclear FOXO1⁺ cells during *in vitro* differentiation (Fig. 6G-M). We observed that the percentages of both ASCL1⁺ and nuclear FOXO1⁺ cells increase early during differentiation (day 2) and then reduce to significantly lower levels (Fig. 6K,L). At the beginning of differentiation, the percentages of ASCL1⁺ and nuclear FOXO1⁺ cells were higher than at the same stage during mouse NEP differentiation (Fig. 6K,L, compared to Fig. 2R,S). Interestingly, the highest expression of ASCL1, nuclear FOXO1, and nuclear FOXO1 and ASCL1 double-positive cells overlapped at day 2, unlike what was observed during mouse NEP differentiation (Fig. 6K-M). In line with this observation, we also detected an 8.69±3.17-fold ($n$=9) increase in the percentage of nuclear FOXO1 and ASCL1 double-

positive cells on day 2 of hNEP differentiation compared to mouse NEPs (Fig. 6N). In summary, while ASCL1 and FOXO1 are transiently upregulated during differentiation of hNEPs, the transient ASCL1[+] and nuclear FOXO1[+] double-positive state seems to be prolonged in hNEP differentiation *in vitro* compared to what is observed in mice.

Finally, to test whether WNT signalling promotes ASCL1[+] to FOXO1[+] transition during *in vitro* hNEP differentiation, we treated the differentiating hNEPs with 1 μM CHIR (Fig. 7A). Activation of WNT signalling led to a significant 1.60±0.34-fold (*n*=5) increase in nuclear FOXO1[+] at day 4 of differentiation, while the percentage of ASCL1[+] cells did not change (Fig. 7B-E), consistent with our observations in mouse NEPs. Although not significant, there was a trend towards an increase in the proportion of nuclear FOXO1 and ASCL1 double-positive cells upon WNT activation at day 4 of differentiation (Fig. 7F). Importantly, activation of WNT signalling during *in vitro* differentiation of hNEPs also increased the percentage of TUJ1[+] cells at day 10 of differentiation 1.57±0.24-fold (*n*=5) compared to the DMSO controls (Fig. 7G-I). Collectively, these data suggest that the ASCL1-WNT-FOXO1 axis during NEP-to-inhibitory neuron differentiation is conserved between mice and humans, while the dynamics of the transient FOXO1[+] state may differ between the species.

## DISCUSSION

Resolving differentiation programmes is crucial for understanding how the nervous system is built in a region-specific manner. This is particularly important where imbalances in one neural type could lead to devastating defects in brain physiology. For example, the

molecular layer inhibitory neurons of the cerebellum play a crucial role in modulating Purkinje cell firing and, hence, cerebellar function (Brown et al., 2019). In addition, molecular layer inhibitory neuron numbers are affected in cerebellar disorders such as ataxias (Edamakanti et al., 2018). However, the molecular mechanisms that regulate cerebellar molecular layer inhibitory neuron differentiation remain poorly understood. In this study, we established previously unreported aspects of the GRN that promote NEP-to-inhibitory neuron differentiation during postnatal cerebellar development using a combination of *in vivo* and *in vitro* datasets and experimental models that include primary mouse and human cerebellar neurosphere cultures. We analysed scRNA-seq of freshly isolated NEPs from the postnatal mouse cerebellum to identify core genes in the GRNs that promote gliogenic and neurogenic differentiation of WM NEPs. We found that WM *Ascl1*-NEPs utilise a combinatorial transcriptional network and resolved the context-dependent functions for ASCL1 and FOXO1, two key neurogenic transcription factors. Our targeted DamID data generated during *in vitro* differentiation suggest that both ASCL1 and FOXO1 promote neurogenesis with temporally dynamic, shared and divergent functions. Both transcription factors facilitate a neurogenic programme, while ASCL1 independently regulates proliferation and FOXO1 expression. FOXO1, on the other hand, promotes survival during NEP differentiation separately from ASCL1. Our *in vitro* data also suggests that WNT signalling induces NEP-to-inhibitory neuron differentiation by facilitating the transition of the *Ascl1*-NEPs from an ASCL1[+] state to a transient FOXO1[+] cell state. Using primary hNEP cultures generated from fetal human cerebella, we demonstrated that the ASCL1-FOXO1 differentiation axis and the role of WNT signalling are conserved during

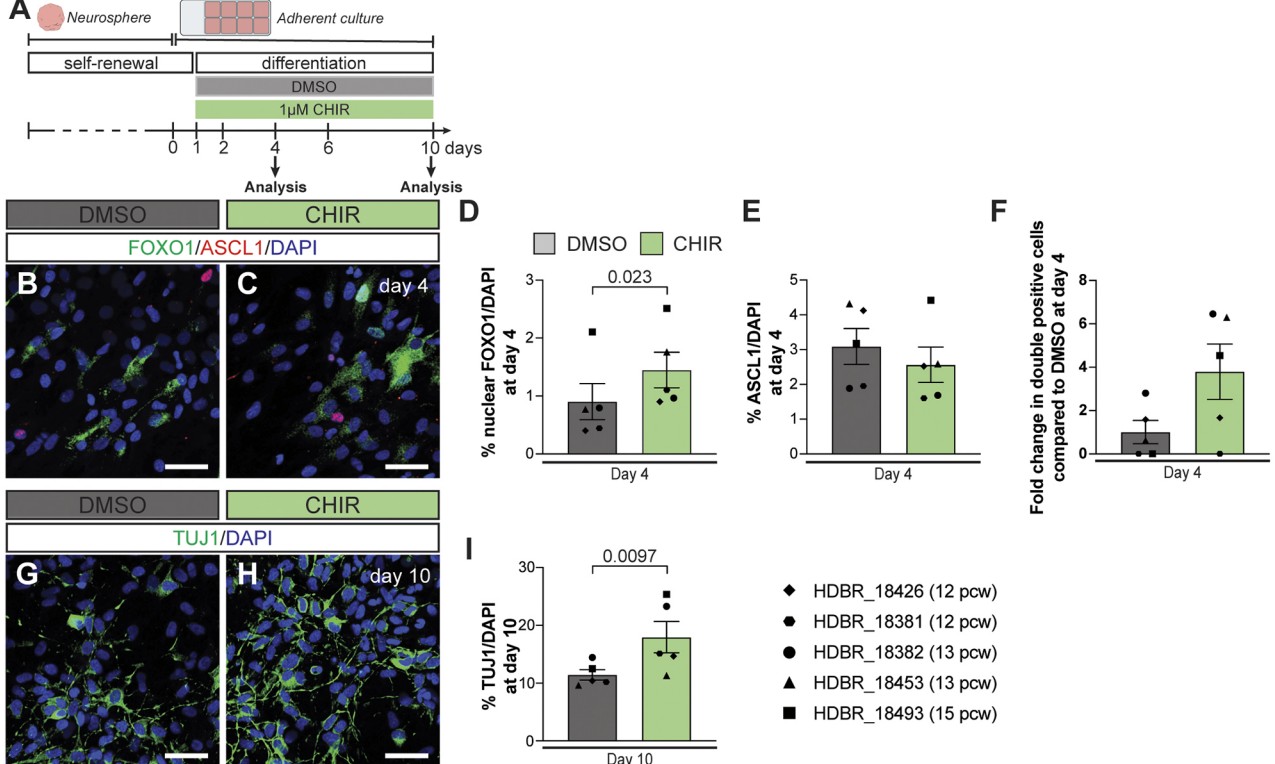

**Fig. 7. The role of WNT signalling in promoting neural production via FOXO1 is conserved in human NEP differentiation.** (A) Experimental design. (B-E) Immunofluorescent analysis (B,C), and quantification of nuclear FOXO1[+] (D) and ASCL1[+] (E) cells in hNEP cultures treated with DMSO or CHIR at day 4 of differentiation (*n*=5, paired Student's *t*-test). (F) Fold change in % nuclear FOXO1 and ASCL1 double-positive cells in cultures treated with CHIR compared to DMSO (*n*=5, paired Student's *t*-test). (G-I) Immunofluorescent analysis and quantification of TUJ1[+] cells in hNEP cultures treated with DMSO or CHIR at day 10 of differentiation (*n*=5, paired Student's *t*-test). Representative images are shown. Data are mean±s.e.m. Scale bars: 50 μm.

differentiation. Based on this work, we propose a model for the role of FOXO1 and its interaction with ASCL1 during NEP-to-inhibitory neuron differentiation in the developing cerebellum (Fig. 8). This work provides unique insights into how cerebellar inhibitory neuron production is regulated while resolving context-dependent functions of ASCL1 and FOXO1 in mouse and human NEPs.

The function of transcription factors is highly context dependent, which has allowed them to be evolutionarily re-used across different cell types and developmental stages (Argelaguet et al., 2022 preprint). Using lentiviral overexpression in primary NEP cultures, we demonstrated that sustained ASCL1 and FOXO1 overexpression in primary NEPs increases neuron production in vitro (Fig. 3J). The sustained overexpression may have disrupted some of the GRNs driving NEP differentiation and could lead to impaired differentiation and maturation. Additionally, our lentiviral overexpression approach is not cell-type specific, resulting in ASCL1 and FOXO1 overexpression in different NEP subtypes in vitro. Therefore, some of the observed differences between ASCL1 and FOXO1 overexpression, such as the differences in the morphology of TUJ1+ cells, could be attributed to these technical challenges. In silico GRN reconstruction of the scRNA-seq data suggested a low level of FOXO1 regulon activity in bipotent NEPs and astrocytes. The role of FOXO1 in these cell types remains to be determined. FOXO1 has been reported to regulate cellular metabolism and inhibit ROS generation in astrocytes in other brain regions (Doan et al., 2023; Wang et al., 2022). Thus, we can speculate that the function of FOXO1 may be partially conserved between cerebellar interneurons and astrocytes.

Our data show that ASCL1 and FOXO1 bind to a shared group of genes related to neurogenesis during in vitro NEP-to-inhibitory neuron differentiation while independently regulating proliferation and cell survival, respectively, at different stages. These observations are consistent with the function of ASCL1 and FOXO1 in other cell types. Additionally, other FOXO family transcription factors have been shown to functionally interact with ASCL1 at the same target genes (Bastie et al., 2005; Brunet et al., 2004; Castro et al., 2006; De La Torre-Ubieta et al., 2010; Imayoshi et al., 2013; Park et al., 2017; Webb et al., 2013). However, it remains unclear whether ASCL1 and FOXO1 are synergistic and/or antagonistic at loci they both bind.

Analysis of Ascl1 transcript levels during NEP-to-inhibitory neuron differentiation shows broad Ascl1 expression, continuing after the Foxo1+ state. Interestingly, immunofluorescent analysis of ASCL1 and FOXO1 levels during in vitro differentiation suggests that ASCL1 protein expression is temporally more restricted. In addition, the rarity of the nuclear FOXO1 and ASCL1 double-positive cells suggests a nontranscriptional control of ASCL1 protein expression. Previous research suggested phosphorylation as key mechanism to control ASCL1 protein expression and function (Ali et al., 2014; Azzarelli et al., 2022). To what extent such mechanisms regulate ASCL1 and their function during NEP-to-inhibitory neuron differentiation remains to be studied. Importantly, while our analysis suggests that the ASCL1 and FOXO1 double-positive state is transient, how the exit from this cellular state is regulated in vitro and in vivo remains to be understood. Perhaps, as the progeny of Ascl1-NEPs migrate away from the niche in the lobule white matter towards the ML, changes in the cellular microenvironment could drive the transitory states. Further in vivo analysis is needed to dissect the role of niche-driven signalling mechanisms during lineage progression.

Cell2Fate analysis revealed dynamic regulation of WNT signalling during NEP-to-inhibitory neuron differentiation (Fig. 5C-H). We demonstrated that CHIR treatment during in vitro mouse and human NEP differentiation increases nuclear FOXO1+ cells and neuron production while not affecting the frequency of ASCL1+ cells (Figs 5J-Q and 7A-I). Similarly, overexpression of ASCL1 during in vitro differentiation resulted in an increased amount of nuclear FOXO1+ cells and neuron production (Fig. 3F,G). The mechanisms underlying how ASCL1 and WNT signalling converge to increase nuclear FOXO1+ levels remain to be investigated. ASCL1 has been shown to promote WNT signalling by directly repressing DKK1, a negative regulator of WNT signalling, and synergising with WNT3A to induce active WNT signalling in glioblastoma stem cells (Rheinbay et al., 2013). Thus, ASCL1 may prime Ascl1-NEPs for WNT signalling. In other systems, WNT signalling has been shown to both positively and negatively regulate FOXO1 in a context-dependent manner (Cheng et al., 2024; Okada et al., 2015; Sreekumar et al., 2017). In a conditional knockout mouse using Nes-Cre to delete

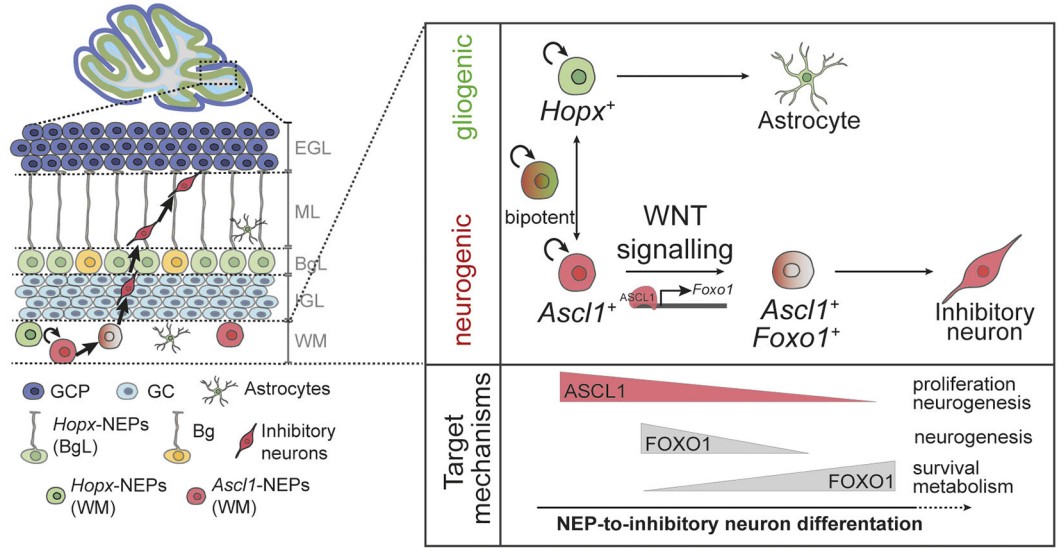

**Fig. 8. Working model for NEP-to-inhibitory neuron differentiation in the developing cerebellum.** Ascl1-NEPs transition to a FOXO1+ state via WNT signalling activation to drive neurogenesis. The downstream targets of ASCL1 and FOXO1 during cerebellar molecular layer inhibitory neuron differentiation provide insights into their context-dependent functions.

WNT5a, a non-canonical WNT ligand, the production of molecular layer interneurons in the postnatal cerebellum was decreased (Subashini et al., 2017). The significant but modest increase in nuclear FOXO1 levels following CHIR treatment *in vitro* further suggests involvement of noncanonical WNT signalling. How the canonical and noncanonical WNT signalling crosstalk promotes NEP-to-inhibitory interneuron differentiation in the postnatal cerebellum remains to be determined. Interestingly, activation of WNT signalling in cerebellar progenitors during embryonic development causes abnormal cerebellar development and a reduced number of inhibitory neurons (Pei et al., 2012; Yang et al., 2019). This highlights the multifaceted role of WNT signalling in cerebellar development, with possibly distinct roles in embryonic and postnatal developmental stages. Furthermore, a downregulation of WNT-related genes was observed after the *Foxo1*+ state in the WM NEP transcriptomes (Fig. 5E-H). FOXO1 has been reported to suppress WNT signalling in neural stem cells (Paik et al., 2009) and other cell types (Guan et al., 2015; Iyer et al., 2013). It is plausible that WNT signalling and FOXO1 form a negative-feedback loop as part of the postnatal NEP-to-inhibitory neuron differentiation programme.

To establish to what extent these mechanisms are conserved in human cerebellum development, we adapted our *in vitro* differentiation paradigm to establish primary hNEP cultures from fetal cerebella. We found that the ASCL1-FOXO1 axis and the role of WNT signalling are conserved in hNEP-to-inhibitory neuron differentiation (Figs 6A-N and 7A-I). Interestingly, we observed differences in the temporal dynamics of ASCL1 and FOXO1 compared to the mouse NEP differentiation, where the proportion of nuclear FOXO1 and ASCL1 double-positive cells was significantly higher during hNEP differentiation (Fig. 6N). We speculate that the prolonged double-positive state may be an evolutionary adaptation to scale up the production of molecular layer interneurons to support the expanded human cerebellar hemispheres (Kebschull et al., 2020).

Finally, our analysis was performed on samples obtained from a wide range of ages (9-17 pcw). We did not find any correlation with the age of the donor. However, it remains unclear whether different cell states or cell types are captured at different ages and how this might influence *in vitro* differentiation. Advancements in human pluripotent stem cell-derived cerebellar organoids may offer an alternative platform to functionally test the molecular mechanisms driving hNEP differentiation (Atamian et al., 2024; Brás et al., 2022). Collectively, our work uncovers parts of the GRN driving NEP-to-molecular layer inhibitory differentiation in mice and humans during late cerebellar development. These findings provide insights into how neurogenesis is regulated in a neuron type- and region-specific manner by resolving the context-dependent functions of ASCL1 and FOXO1. Our results have the potential to inform future therapies for disorders where inhibitory neuron production is impaired.

### Limitations

Our *in vitro* primary neurosphere cultures are a powerful tool for identifying molecular mechanisms of NEP behaviour in mice and humans. Whether primary NEP cultures retain their lineage propensities *in vivo* following orthotopic injections remains to be determined. Importantly, these findings should be validated using CKO mouse models for FOXO1 during postnatal cerebellum development. Targeted DamID results require further validation via approaches that test native protein binding. However, the low number of relevant cells (*Ascl1*-NEPs) that can be isolated from the neonatal mouse brain and the low frequency and the transient nature

of the ASCL1+ and FOXO1+ NEP states *in vivo* and *in vitro* limit such studies. We restricted our targeted DamID analysis to regions within 2 kb of the TSS, which neglected distal regulatory elements. Finally, the limited availability of human fetal tissue, particularly from later stages, and the inability to maintain cultures long-term pose a significant challenge to characterising developmental dynamics and performing functional validation in hNEPs.

## MATERIALS AND METHODS

### Animal husbandry

All the mouse experiments were performed according to the protocols approved by Home Office Project Licence PP8035780. The mouse lines used in this study are: *Nes-CFP* (Mignone et al., 2004; Wojcinski et al., 2017), *Ascl1*$^{CreERT2}$ (Pacary et al., 2011; Sudarov et al., 2011), *Rosa26*$^{lox-stop-lox-TdTomato}$ (ai14, stock 007909, The Jackson Laboratory) (Takeda et al., 2011) and *Ascl1*$^{fl/fl}$ (Pacary et al., 2011; Sudarov et al., 2011). Animals were maintained on an outbred *Swiss Webster* or *CD1* background, housed on a 12-h light/dark cycle and had access to food and water *ad libitum*. Both sexes were used for the study.

For genetic inducible fate mapping, P1 pups were injected subcutaneously with 200 µg/g tamoxifen (Sigma-Aldrich, T5648). For ASCL1 CKO, pups were given 200 µg/g tamoxifen at P0. For proliferation analysis, 50 µg/g EdU (Thermo Fisher, E10187) was injected subcutaneously 1 h before the experimental endpoint.

### Human fetal tissue

9-17 pcw old human fetal cerebellar tissue was obtained from either the Cambridge University Hospitals NHS Foundation Trust under permission from NHS Research Ethical Committee (96/085) or the MRC/Wellcome Trust Human Developmental Biology Resource London [University College London (UCL) site REC reference: 18/LO/0822] and Newcastle (Newcastle site REC reference: 18/NE/0290), project 200702 (www.hdbr.org). Only samples with no genetic abnormalities were used.

### Primary NEP neurosphere cultures

Cerebella from P1 *Nes-Cfp/+* or *Swiss Webster* pups were dissected and dissociated using Accutase (Thermo Fisher, A1110501) for 10 min at 37°C. Once a single cell suspension was obtained, cells were washed with NSC media [Neurobasal (Thermo Fisher, 10888022) with 1×B27 (Thermo Fisher, A3353501), 1×N2 (Thermo Fisher, 11520536), 2 mM L-glutamine (Thermo Fisher, A2916801), 1× non-essential amino acids (Thermo Fisher, 11140035) and 100 U/ml penicillin-streptomycin (Thermo Fisher, 15140122)]. For cultures generated from *Nes-Cfp/+* cerebella, CFP+ cells were FACS isolated. The *Swiss Webster* cells were directly plated to form neurospheres. Initial cell suspensions were plated onto ultra-low attachment plates ~100,000 cells/ml density. Cultures were maintained with 20 ng/ml of EGF (Fisher Scientific, PMG8041) and FGF2 (Qkine, Qk042-0500).

Primary hNEP neurospheres were established similarly to the mouse NEPs generated from the *Swiss Webster* pups, except for an additional filtering step with a 40 µM nylon filter (Merck, CLS352340) after dissociation to remove debris. hNEP cultures were maintained in the NSC media supplemented with human EGF (QKINE, Qk011-0100) and FGF2 (QKINE, Qk027-0100). Growth factors were supplemented every other day, and media were changed weekly. Primary cultures with <20 passages were used for studies.

### *In vitro* differentiation

Chamber slides (Thistle Scientific, IB-80806) were coated with poly-D-lysine and laminin. Briefly, eight-well slides were incubated with 250 µl poly-D-lysine (Thermo Fisher, A389040) for >2 h at 37°C. The plates were rinsed with PBS+/+ (+Mg and +Ca, Thermo Fisher, 14040091) twice and then were incubated with 2 µg/cm² laminin (Merck, L2020) or human laminin-521 (Thermo Fisher, A29249) in PBS+/+ (Thermo Fisher, 14040091). The plates were stored at 4°C until use. On the day of differentiation, the plates were incubated for >1 h at 37°C and washed once with PBS+/+ prior to cell seeding.

Neurospheres were dissociated using Accutase and washed with NSC media once. Cells were plated at a density of 250,000 cells/cm$^2$ (mouse) or 100,000 cells/cm$^2$ (human) on poly-D-lysine and laminin-coated plates. Cells were kept in self-renewing conditions (NSC media+growth factors) overnight and then switched to differentiation media [NSC media+2% fetal bovine serum (FBS, Thermo Fisher, 10438026)]. The media were changed every other day and on days before fixation. For WNT activation, cells were treated with 1 µM CHIR99021 (BioTechne, 4423/10). 0.1% DMSO was used as a control. HDBR_18426 (12 pcw), HDBR_18381 (12 pcw), HDBR_18453 (13 pcw), HDBR_18382 (13 pcw) and HDBR_18493 (15 pcw) were used in the human WNT experiments. Two or three technical replicates were performed for each sample and averaged.

Cells were fixed at room temperature [15 min for 2D cultures and 30 min for neurospheres using 4% paraformaldehyde (PFA, Thermo Fisher Scientific, 043368.9M)] and washed thrice with 1×PBS. Fixed cells were stored at 4°C until immunofluorescent analysis.

### Tissue preparation for histological analysis
The dissected neonatal mice brains or fetal human cerebellum were drop fixed in 4% PFA at 4°C for up to 1 week followed by 30% sucrose (Merck, 1076511000) in PBS until the brains sunk. The brains were then embedded in OCT (VWR, 361603E) blocks and stored at −80°C until further processing. 14 µm sections were obtained using a cryostat (Leica CM3050S) and stored at −80°C.

### Immunofluorescent analysis
Tissue sections, neurospheres and 2D cultures were subjected to immunofluorescent analysis. The list of antibodies used for this analysis is provided in Table S8. Slides were allowed to come to room temperature. Samples were blocked with blocking buffer [5% bovine serum albumin (Merck Life Science, A9418-50G) in 1×PBS and 0.1% Triton-X)]. For ASCL1 staining on slides, mouse-on-mouse blocking was performed according to the manufacturer's guidelines (Thermo Fisher, R37621). Primary antibody incubation was performed in blocking buffer overnight at 4°C. Samples were then washed three times with 1×PBS with 0.1% Triton-X for 5 min. Secondary antibody incubation was performed in blocking buffer for 1-2 h at room temperature, and samples were protected from light for the rest of the procedure. After secondary antibody staining, cells were washed three times with 1×PBS with 0.1% Triton-X for 5 min. DAPI (Sigma-Aldrich, MBD0015-1ML) was used for counterstaining. Sections were mounted with fluoro-gel (EMS, 17985-10). Cultures were stored in 1× PBS. Cells and sections were stored at 4°C in the dark until imaging.

A similar procedure was used for neurosphere staining with minor modifications to the blocking buffer [PBS containing 0.2% bovine serum albumin, 0.02% sodium dodecyl sulfate (SDS) and 0.3% Triton X-100] and extended incubation times.

Apoptosis was detected using TUNEL staining according to the manufacturer's guidelines (Thermo Fisher, 16314015). Where relevant, EdU was detected using Click-it assay with sulfo-azide-Cy5 (Lumiprobe, A3330).

### *In situ* hybridisation chain reaction
Gene-specific probes and specific reagents were ordered from Molecular Instruments and the manufacturer's instructions were followed. Briefly, cryosections were equilibrated to 4°C and postfixed for 15 min at 4°C in 4% PFA. Thereafter, sections were dehydrated by sequential 5 min washes in 50%, 75% and 100% ethanol, and washed in 1× PBS thrice. Sections were incubated with 200 µl preheated *In situ* hybridisation chain reaction (HCR) Probe Hybridization Buffer at 37°C in a preheated NeoBrite (NeoBiotech) humidified chamber for 10 min. 2 µl of each probe was diluted in 300 µl HCR Probe Hybridization Buffer and incubated with the sections overnight at 37°C in the humidified chamber. Sections were then washed four times in preheated HCR Probe Hybridization Buffer for 15 min at 37°C and then incubated with ambient HCR Gold Amplifier Buffer at room temperature for 30 min. For each reaction, 2 µl of hairpins H1 and H2 targeting the relevant adapters were snap-cooled by incubating at 95°C for 90 s followed by 30 min at 10°C in a thermocycler (Eppendorf Mastercycler X50a). Hairpins were diluted in 300 µl HCR Gold Amplifier Buffer per reaction and incubated with the sections overnight at room temperature in a humidified chamber. The sections were then washed four times in HCR Probe Wash Buffer for 15 min supplemented with DAPI (1:1000) in the second and third washes, and mounted with fluoro-gel (EMS, 17985-10). The sections were stored at 4°C in the dark until imaging.

### Image acquisition and analysis
Images were acquired using a Leica CKX53 epifluorescence microscope, or a Zeiss LSM880 confocal or Andor BC43 CF confocal microscope. Images were processed using ImageJ. Images within an experiment were acquired with identical settings.

Quantification of tissue sections was performed on lobules 4 and 5 in three midsagittal sections/brain. Quantification was carried out manually using ImageJ/FIJI Cell Counter Plugin (NIH). For experiments that involved 2D cultures, three images were acquired from each well and 3-10 wells/condition were quantified across >3 individual experiments. Quantification was carried out manually using ImageJ/FIJI Cell Counter Plugin (NIH). TUNEL quantification was carried out with FIJI, converting the images to an 8-bit image, and quantifying particles with the Analyze Particles tool.

### Lentiviral production
VSV-G pseudotyped lentivirus was produced according to standard protocols using HEK293T cells, and the packaging plasmids for Gag/pol, VSV-G, REV and Tat plasmids (ratio 1:2:1:1) were added together with 5 µg of transfer plasmid. Upon collection, the media were filtered through a 0.2 µm filter (Merck, SLGSVR255F) and concentrated using Lenti-X reagents (Takara, 631232) according to the manufacturer's guidelines. The concentrated lentivirus was resuspended in 300 µl of Opti-MEM (Thermo Fisher, 11058021) per 10 cm dish. The virus was aliquoted in appropriate volumes, snap-frozen on dry ice and stored at −80°C until use. Viral titers were calculated using ELISA. The transfer vectors used in this study are provided in Table S10.

### Lentiviral infection *in vitro*
Primary NEP neurospheres were dissociated into single cells as described above and incubated with the respective virus at 1-10 multiplicity of infection (MOI) and 4 µg/ml protamine sulfate (Merck, 1101230005) for 12-18 h.

For the overexpression experiments, a stable rtTA NEPs cell line was first established and then reinfected with the subsequent overexpression virus prior to analysis. These cells were differentiated 3-10 days after infection as described above, with the exception that Tet system approved FBS (Thermo Fisher Scientific, A4736401) was used in the differentiation media. In the doxycycline condition, 1 µg/ml doxycycline (Sigma, D9891) was added to the differentiation media for days 1-3 (Fig. 3A).

shRNAs were designed using the Broad Institute TRC shRNA design tool. For shRNA mediated knockdown, four different *Foxo1* shRNAs (*Foxo1*-sh1, CCGCCAAACACCAGTCTAAAT; *Foxo1*-sh2, CGGAG-GATTGAACCAGTATAA; *Foxo1*-sh3, TGTAATGATGGGCCCTAAT-TC; *Foxo1*-sh4, CATGGACAACAACAGTAAATT) and one *Ascl1* shRNA (*Ascl1*-sh1, CAAGTTGGTCAACCTGGGTTT) and a scrambled shRNA oligo were cloned into the pLKO.1 vector as previously described (Moffat et al., 2006). NEPs were infected and selected with 1 µg/ml puromycin for 1 week to establish stable cell lines. Following infection, *Foxo1* expression levels were assessed by qPCR or immunofluorescence and 2/4 shRNAs (*Foxo1*-shRNA1 and *Foxo1*-shRNA2) that showed better knockdown were carried on for further analysis (Fig. 3L). ASCL1 levels were assessed via immunofluorescent staining (Fig. S5B).

### Targeted DamID
FOXO1 or ASCL1 CDS was inserted into the SFFV-mNeonGreen-Dam lentiviral vector (Lim et al., 2023) by Gibson assembly. Differentiations were performed in 6-well plates (Thermo Scientific, 140675). Differentiating NEPs were infected with Dam-alone, ASCL1-DAM or FOXO1-Dam lentivirus in neural stem cell media with 2% fetal bovine serum and 4 µg/ml protamine sulfate, and were incubated at 37°C for 6 h. Following infection, the media was changed to differentiation media and cells were collected for downstream analysis after 24 h (Fig. 4A). Briefly, cells were dissociated with Accutase and

washed once, and pellets were collected and stored at −80°C until library preparation.

Genomic DNA was extracted from cell pellets and processed as previously described (Marshall et al., 2016). Sequencing was performed by the Gurdon Institute NGS core as single end 100 bp reads on an Illumina NovaSeq 6000. Trimmed fastq files were quality checked and validated using FastQC (v0.12.0) then analysed with a modified version of the damidseq_pipeline (Marshall and Brand, 2015), damMer. DamMer generates normalised binding tracks and identifies statistically significant and reproducible peaks (Tang et al., 2022). Briefly, reads were mapped to mm10 genome assembly, indexed using Bowtie2 (v2.3.4.1) (Langmead and Salzberg, 2012) and binned into 5′-GATC-3′ fragments. Each fusion protein (FOXO1/ASCL1) sample was normalised individually against each replicate of Dam-alone control treatment, then averaged using RPM normalisation, in 300 bp bins. Binding intensity values were quantile normalised across all replicates for each stage and backtransformed. Files were converted to bigwig format using bedGraphToBigWig (v4) and imported into the Integrative Genomics Viewer (IGV v2.17.4) for visualisation. Macs2 (Zhang et al., 2008) (v2.1.2) was used to call broad peaks for fusion and Dam-alone pairs using the bam file output of the damidseq_pipeline, retaining a broadpeak file for each Dam-alone replicate for analysis of chromatin accessibility. Overlapping peaks were merged using bedtools (v2.26.0) to generate consensus peak files for fusions and Dam-alone files. The fusion protein peaks were called after normalisation to the Dam-alone control, while Dam-alone data was not normalised before peak calling. All peaks were then filtered by a stringent FDR-threshold of $1{\times}10^{-5}$ to identify significant binding peaks. For downstream analyses, significant peaks were further filtered to those present only in 3/4 replicates (repropeaks) using bedtools (Quinlan and Hall, 2010) (v2.26.0), denoted as reproducible peaks.

The repropeak bed files were read into R using the ChIPpeakAnno (Zhu et al., 2010) (v3.32.0) function toGRanges() with format=‘BED’, header=T and otherwise default parameters. The Grange objects were annotated individually using the function annotatePeakInBatch() with AnnotationData=genes (EnsDb.Mmusculus.v79). Peaks were filtered based on their proximity to the nearest TSS, retaining only those located within ±1 or 2 kb. The feature list was then made unique. A list of the genes with a significant peak ±1 or 2 kb of the transcriptional start site was made and used to compute unique and shared genes using ggven (v0.1.14) with default parameters. A conversion table was made of ENSEMBL IDs and gene symbols using BiomaRt (Smedley et al., 2009) (v2.54.1) getBM() specifying attributes=c('ensembl_gene_id', 'ensembl_transcript_id', 'mgi_symbol') and mart=useMart(biomart="ENSEMBL_MART_ENSEMBL", dataset="mmusculus_gene_ensembl", host="https://www.ensembl.org/", verbose=F). The gene set was exported as a csv-file using write.csv() with default parameters. GO term analysis was performed using PANTHER (Thomas et al., 2022) (v19.0) via the online GUI available at https://pantherdb.org/. GO term analysis for biological process, molecular function and cellular component terms was performed individually with default parameters using Fisher's exact test and a false discovery rate calculation to correct for multiple testing. For individual transcription factors, all genes with a reproducible significant peaks ±1 or 2 kb of the transcriptional start site were used for the analysis, irrespective of the other transcription factor binding. For the co-bound analysis, genes must have reproducible significant peaks ±1 or 2 kb from the transcriptional start site for both transcription factors at the given timepoint. Overlapping peaks were identified using findOverlapsOfPeaks() with default parameters and minoverlap=1.

Motif enrichment analysis was performed on ASCL1-Dam, FOXO1-Dam and Dam-alone datasets at day 2 and day 6 of differentiation. All computed significant peaks were used for the analysis (Fig. 4B, Table S4). The produced bed-files were loaded using toGRanges() and annotated as described above. From the Grange object, a HOMER-compatible data frame was constructed. The HOMER-compatible data frame was exported as a bed-file using write.table() specifying quote=F, sep="\t", row.names=F, col.names=F. Motif enrichment analysis was then performed using the HOMER (Heinz et al., 2010) (v5.0.1) function findMotifsGenome.pl using the mm10 genome specifying -mask and otherwise default parameters. Results from the knownResults output were reported, and multiple testing correction (Bonferroni's) was applied. The enrichment score was calculated as the ratio of a motif's frequency in target sequences versus background sequences, sampled by HOMER using default parameters. An enrichment score above 1 means that the motif occurs more often in target sequences than in the background.

Finally, to identify potential DNA-binding proteins that are enriched within the shared genes that showed binding of both ASCL1-Dam and FOXO1-Dam, we took on a different approach. Shared genes containing reproducible (3/4 replicates) ASCL1-Dam and FOXO1-Dam peaks, either ±1 kb or 2 kb of their TSS, were identified using the intersect() function. Next, the reproducible (3/4 replicates) Dam-alone peaks were annotated to their nearest gene. This GRange object was then subsetted to contain only the peaks near the genes that had been identified to have reproducible (3/4 replicates) ASCL1-Dam and FOXO1-Dam peaks (repeated for both ±1 and 2 kb of their TSS). These reproducible Dam-alone peaks were then used for the HOMER analysis. The analysis was repeated for both day 2 and day 6 data (Table S4, tabs designated as HOMER_Dam-alone_shared_1/2kb_day2/6).

## Single cell RNA-sequencing and data analysis

### Sample preparation, scRNA-seq, sequencing and data processing

Nes-CFP⁺ cells isolated from postnatal mouse cerebella (in vivo mouse dataset) Sample preparation, cell multiplexing and droplet preparation, and mapping and initial quality control were performed as described by Pakula et al. (2025). NonIR (control condition in Pakula et al., 2025) cells from P1-3 and 5 cerebella were subsetted from the Seurat object using Seurat (Hao et al., 2024) (v4.1). The object was then split into a list based on the individual biological replicates using SplitObject, and normalised using ScaleData() with default parameters. The list of Seurat object was prepared for integration using SelectIntegrationFeatures() and PrepSCTIntegration() using default parameters, and subsequently integrated using IntegrateData() with default parameters. The integrated Seurat object was normalised using SCTransform() with vars.to.regress=c("percent.mt", "CC.Difference", "nFeature_RNA", "nCount_RNA"). Dimension reduction was performed using RunPCA() using all features in the integrated assay and npcs=100 and RunUMAP() with parameters dims=1:40, seed.use=888, repulsion.strength=0.1 and min.dist=0.5 in addition to default parameters. The clustering analysis was performed as described below.

Primary mouse and human NEP cultures (in vitro mouse and human datasets) The different primary neurosphere lines were dissociated separately using Accutase and washed once in NSC media as described above. The cell suspensions were filtered and pelleted by centrifugation at 500 ***g*** for 5 min at room temperature and washed once in PBS+0.04% bovine serum albumin. Samples (2 primary mouse NEP lines and 10 primary hNEP lines) were then multiplexed using CellPlex (10x Genomics) oligos using the manufacturer's instructions. After the final wash, different samples were pooled at a concentration of 1600 cells/µl to target 2500 cells/sample. A total of 30,000 pooled cells were targeted for droplet formation using Chromium X/iX (10X Genomics) according to the manufacturer's guidelines and the Next-GEM 3′ v3.1 chemistry. Library preparation was performed according to manufacturer's guidelines by the Cambridge Stem Cell Institute NGS Core Facility, and the sequencing was performed on a NovaSeqX at the Cancer Research UK Cambridge Institute Genomics Core Facility. The data were aligned using CellRanger (v8.0.1) multi with default parameters, aligning mouse data to GRCm39 with GENCODE vM33/Ensembl110 annotations and human data to GRCh38 with GENCODE v44/Ensembl110 annotations.

A Seurat object was created from the CellRanger output using the Read10X() function (Seurat v5.2.1). Cells assigned as either "Multiplet" or "Unassigned" in the assignment confidence table, or having 0 reads in the CMO (CellPlex) assay were filtered out using the subset() function. Demultiplexing was performed via the Seurat NormalizeData() function with normalization.method="CLR" and default parameters followed by HTODemux() with assay="CMO", positive.quantile=0.99 and default parameters. The data was then subsetted based on the hash.ID using the subset() function. Additionally, for the primary hNEP neurosphere dataset, the velocyto (v0.17.17) function run() was run on the CellRanger

output with –dtype uint32, the GRCh38 reference genome and otherwise default parameters. A Seurat object was constructed from the output using the CreateSeuratObject() function with the parameters min.cells=0, min.features=0, names.field=2, names.delim="\\-", adding both a spliced and unspliced assay. The Seurat Object constructed from the CellRanger output was then subsetted to only contain cells, which contains spliced and unspliced information. Quality control of mouse cells was carried out using the subset() function, specifying nFeature_RNA>200 & nFeature_RNA<7500 & percent.mt<7.5. Quality control of human cells was similarly carried out with the subset() function, specifying nFeature_RNA>200 & nFeature_RNA<7500 & percent.mt<5. The clustering analysis was performed as described below.

### 17 and 20 pcw human fetal cerebellum

Single nucleus RNA-sequencing data described by Sepp et al. (2024) were obtained from the heiDATA repository (https://doi.org/10.11588/data/QDOC4E) for the samples (SN120, SN121, SN133, SN297, SN136, SN232 and SN308 for the time points 9 wpc, 11 wpc, 17 wpc, 20 wpc, 7 months, CS19 and newborn, respectively). Raw snRNA-seq data were aligned to the Human reference genome GRCh38 using STARsolo/STAR (v2.7.3a) (Kaminow et al., 2021 preprint). In addition to the default parameters, the following parameters were adjusted to optimise the mapping of transposable elements using STARsolo "–outFilterMultimapNmax 100 –winAnchorMultimapNmax 100 –outSAMmultNmax 1" and the following parameters were included to facilitate subsequent analyses: "–outSAMattributes NH HI AS nM CR CY UR UY GN CB UB GX GN sS sQ sM –soloFeatures Gene Velocyto". Repeatmasker file for the GRCh38 genome was obtained from UCSC and converted to the required format as described in the SoloTE (v1.10) (Rodríguez-Quiroz and Valdebenito-Maturana, 2022) documentation. The SoloTE pipeline with default parameters was then run using the repeat annotation file and alignment files (in the sorted bam format) for each of the samples. SoloTE outputs were read into Seurat using ReadMtx() function, cells were filtered using the subset function based on the following criteria (nFeature_RNA>200 & nFeature_RNA<6000 & percent.mt<10), and a Seurat object including all samples was created using CreateSeuratObject().

Cells from 17 and 20 pcw were subsetted using the subset() function, and the soloTE assay was discarded. Cell cycle scoring was performed using the CellCycleScoring() function using the supplied cell cycle genes from Seurat. The difference in S.Score and G2M.Score was calculated and stored as a meta data variable called CC.Difference. Next, the data were normalised using the SCTransform() function with the parameters vars.to.regress=c("percent.mt", "CC.Difference", "nFeature_RNA", "nCount_RNA"). Dimension reduction was performed using RunPCA() with default parameters followed by RunUMAP() with dims=1:30, reduction="pca", return.model=TRUE, seed.use=1234 and default parameters. Due to the presence of low-quality cells, the data was subsetted using the subset() function with nFeature_RNA>500 & nCount_RNA>500. Normalisation and dimension reduction was then re-performed as described using the SCTransform(), RunPCA() and RunUMAP() functions with the specified parameters. Cluster analysis was performed as described below.

### Clustering

Clustering was performed using FindNeighbors with default parameters and dims=1:30 and FindClusters with default parameters and a resolution varying between 0.1 and 3 in 0.1 increments. Candidate clustering resolutions were guided by maximising the silhouette score and minimising the negative silhouette score. Among the candidate resolutions, the biological meaningfulness of the resulting clusters was evaluated by the expression of marker genes known to delineate previously defined cerebellar cell types, NEP subtypes and cellular states (Bayin et al., 2021; Pakula et al., 2025; Sepp et al., 2024). Cluster analysis was carried out using PrepSCTFindMarkers() using the SCT assay and default parameters followed by FindAllMarkers() with the only.pos=TRUE, min.pct=0.10, logfc.threshold=0.25 and assay="SCT". Clusters were annotated based on known marker gene expression and/or histological analysis (such as RNA *in situ* from Allen Brain Atlas), where relevant for cluster markers (Bayin et al., 2021). The clustering and cluster analysis of all

NEPs and white matter subset were performed using the same functions and parameter settings. The white matter subset was exported as a .h5ad object using the functions SaveH5Seurat() and Convert() from SeuratDisk (v0.0.0.9020).

### Integration of *in vitro* and *in vivo* datasets

To map the *in vitro* mouse and human NEP neurosphere cultures to the *in vivo* reference datasets (all Nes-CFP[+] (Fig. 1E) and data of Sepp et al. (2024) (Fig. S8E), the top 200 variable genes of the reference datasets were computed using the VariableFeatures() function with nfeatures=200 and default parameters, and the *in vitro* data was subsetted to contain these genes using base R. When mapping the mouse primary NEP neurospheres to the all NEPs (Fig. 1F), and neurogenic and gliogenic WM NEP subsets (Fig. 1G), all genes were used to better capture the subtle transcriptional diversity between cell states. In the case of the *in vivo* mouse data, the dimension reduction was reperformed using the RunPCA() function with dims=1:30, n.neighbors=30, return.model=T and default parameters. Furthermore, the feature names of the *in vitro mouse* data were made compatible with the *in vivo* data by mapping the gene symbols into ENSEMBL IDs. Changing the feature names was carried out by converting the Seurat object into a SingleCellExperiment object using the as.SingleCellExperiment() function with assay="SCT" and then mapping the feature IDs using the mapIds() function (EnrichmentBrowser v2.2) and org.Mm.eg.db (org.Mm.eg.db v3.16) with keytype="SYMBOL", columns="ENSEMBL", column="ENSEMBL" and default parameters. The SingleCellExperiment object was converted back into a Seurat object using the function as.Seurat(). Integration anchors were computed between the two datasets by the FindTransferAnchors() function with dims=1:40, reference.reduction="pca", normalization.method="SCT" and default parameters. The integration anchors were then used to perform the mapping using the MapQuery() function with reference.reduction="pca", reduction.model="umap", query.dims=1000 and refdata set to the clustering resolution determined previously. Mapped cells with predicted.celltype.score below 0.6 were set to Unassigned. The results of the integrations were visualised using geom_sankey (ggsankey v0.0.99999).

### scVelo

Trajectory analysis was performed using scVelo (Bergen et al., 2020; La Manno et al., 2018) (v0.3.1). Filtering and normalisation were carried out using pp.filter_and_normalize with default parameters, except min_shared_counts=20, *n*_top_genes=5000, and moments calculated using pp.moments with default parameters, except *n*_pcs=30, *n*_neighbors=30. Thereafter, splicing kinetics was computed using tl.recover_dynamics with default parameters, and gene velocities were estimated using tl.velocity with mode='deterministic', use_latent_time=True. The steady-state model was used due to the appearance of gene phase plots. Finally, the velocity graph was computed using tl.velocity_graph with default parameters, except mode_neighbors='connectivities', and terminal states and latent time with tl.terminal_states and scv.tl.latent_time, respectively, with default parameters, without choosing a root cell. Trajectories were visualised using pl.velocity_embedding_stream using clusters generated using Seurat. The computed pseudotime was exported as a csv and imported into R. It was added to the gliogenic and neurogenic NEPs in the white matter Seurat object metadata using the AddMetaData() function. Regression of normalised gene expression (SCT assay) with respect to the pseudotime was carried out using the geom_smooth() function from ggplot2 (v3.5.1) with default parameters in addition to method="loess", se=TRUE and alpha=0.2.

### pySCENIC

*In silico* gene regulatory network reconstruction was performed using pySCENIC (Van de Sande et al., 2020) (v0.12.1). A loom file was produced from the .h5ad output of the Seurat analysis. The GRNBoost2 network inference algorithm grn was run using default parameters and using the mouse or human database of transcription factors for mouse and human data, respectively. Enriched motifs were computed using ctx with default parameters using the mm10 motif ranking database and the v10 annotation database. Regulons were then calculated based on the enriched motifs using df2regulon with default parameters. Finally, regulon enrichment was calculated using aucell using default parameters and regulon specificity

scores were calculated by regulon_specificity_scores with default parameters. Owing to the stochastic nature of the network inference algorithm, the analysis was performed 10 times to ensure reproducible results. All databases used are available here: https://resources.aertslab.org/cistarget/.

## CellOracle

The subset containing gliogenic and neurogenic NEPs from the WM was exported from R into Python, and Scanpy (Wolf et al., 2018) (v1.9.8) was used for pseudotime calculation and the generation of an anndata object. *In silico* transcription factor perturbations were carried out using CellOracle (Kamimoto et al., 2023) (v0.16.0) following the standard pipeline using default settings at all functions (https://morris-lab.github.io/CellOracle.documentation/tutorials/simulation.html) and using CellOracle's pre-computed base GRN from mouse scATAC-seq atlas. Based on the previous computational and experimental analyses, a cell from the middle of cluster 4 was chosen as the root cell. Therefore, the cell TAGACTGTCCAGTTCC was chosen as the root cell to compute pseudotime. The impact of *in silico* knockout and overexpression of candidate transcription factors identified using pySCENIC or from previous literature on cell state transitions was simulated. To simulate the knockout of a given transcription factor, its expression value was set to 0. For overexpression simulations, the expression was set to the maximum expression value of that transcription factor in any cell.

## Cell2Fate

To explore the underlying biology resulting in the predicted cell trajectories, Cell2Fate (v0.1a0) (Aivazidis et al., 2025) was used. The .h5ad object from the Seurat analysis was imported as an anndata object. Training data was chosen using the function c2f.utils.get_training_data() with cells_per_cluster=10**5, cluster_column='SCT_snn_res.0.6', min_shared_counts=20, $n$_var_genes=3000 and otherwise default parameters. The anndata object was registered using scvi-tools function Cell2fate_DynamicalModel.setup_anndata. The maximal number of modules was calculated using c2f.utils.get_max_modules with default parameters, and the model was initialised using c2f.Cell2fate_DynamicalModel with the computed maximum modules and default parameters. Then, the model was trained using the function train with default parameters. The training was validated based on the convergence of the evidence lower bound using view_history(). Following the training, the model was exported using "export_posterior with num_samples": 20, "batch_size": 2000, "use_gpu": False, 'return_samples': True. The module activity was then calculated using compute_module_summary_statistics with default parameters and visualised using plot_module_summary_statistics. Genes enriched in each module were computed and ranked using get_module_top_features using *P*_adj_cutoff=0.01 and all expressed genes as background. The ranked list of the top 200 most enriched genes in each module was subsequently used for GO term analysis. GO term analysis was performed using PANTHER (Thomas et al., 2022) (v19.0) via the online GUI available at https://pantherdb.org/. GO term analysis for biological process, molecular function and cellular component terms was performed individually with default parameters using Fisher's exact test and a false discovery rate calculation to correct for multiple testing.

## Statistical analysis

Prism (GraphPad) was used for statistical analysis. All data are presented as mean±s.e.m. unless otherwise stated. Sample sizes for biological replicates are mentioned in text and figure legends, where relevant. Statistical tests performed for each analysis were mentioned in the figure legends and Table S9. When one- or two-way ANOVA was used, multiple comparisons tests were performed using Tukey's multiple comparisons test, except for Figs 6N (Šídák's) and S3A-H (Benjamini, Kreiger and Yukutieli) based on the Prism's recommendations. Significant *P*-values for multiple comparisons are shown in the figure. All *t*-tests performed were two-tailed. See Table S9 for a summary of all statistics performed.

## Acknowledgements

We thank the members of the Bayin Lab for their constructive feedback, and Dr James Bae and Niamh Divers for editing the manuscript. We thank Steven Lisgo, Nita Solanky and HDBR, and Roger Baker, Xiaoling He and the Cambridge Brain Repair Centre for sharing human fetal tissue. We thank Drs Iva Tchasovnikarova, Emma Rawlins, Julie Ahringer and Lindsay LaFave for sharing resources and feedback. Finally, we thank the UBS Gurdon Institute Animal Facility, the Gurdon Institute Imaging Facilities, the Gurdon Institute Sequencing Facilities, the Gurdon Institute Bioinformatics Group, the Gurdon Institute Scientific Computing Facility, the Cambridge Stem Cell Institute and the Cancer Research UK Cancer Institute Genomics Core for their outstanding technical help.

## Competing interests
The authors declare no competing or financial interests.

## Author contributions
Conceptualization: J.B.C., N.S.B.; Data curation: J.B.C., A.J.R.; Formal analysis: J.B.C., N.S.B.; Funding acquisition: A.H.B., N.S.B.; Investigation: J.B.C., A.P.A.D., M.M., G.V., M.H., J.T.H.L.; Project administration: N.S.B.; Resources: O.A.B., A.H.B.; Supervision: A.J.R., N.S.B.; Visualization: J.B.C.; Writing – original draft: J.B.C., N.S.B.; Writing – review & editing: J.B.C., A.P.A.D., M.M., G.V., M.H., J.T.H.L., O.A.B., A.H.B., N.S.B.

## Funding
This work was funded by the University of Cambridge School of Biological Sciences DTP PhD Studentship and Peter and Emma Thomsen's Scholarship (1051 to J.B.C.), by a Brain Research UK PhD Studentship (PhD23-100024 to G.V.), by Wellcome Trust grants (206194 and 220540/Z/20/A to O.A.B.), by a Wellcome Career Development Award (227294/Z/23/Z to N.S.B.), by the Royal Society (RGS\R1\231143 to N.S.B.), by Cambridge Stem Cell Institute Seed Funding, by a Wellcome Trust Senior Investigator Award (103792 to A.H.B.), by a Wellcome Investigator Award (223111 to A.H.B.), and by a Royal Society Darwin Trust Research Professorship (RP150061 to A.H.B.). The Gurdon Institute was funded by Wellcome Core Grant (203144) and by a Cancer Research UK grant (C6946/A24843). Open Access funding provided by the University of Cambridge. Deposited in PMC for immediate release.

## Data and resource availability
The mouse cerebellum scRNA-seq data have been deposited in ArrayExpress under accession number E-MTAB-13353. The primary mouse and human NEP scRNA-seq datasets have been deposited in Gene Expression Omnibus under accession number GSE301654. The targeted DamID sequencing data is also available in Gene Expression Omnibus (accession number GSE292043). Scripts to process DamID data are available in GitHub: https://github.com/AHBrand-Lab/DamID_scripts. Scripts used for data analysis are available in GitHub: https://github.com/BayinLab/Christensen_et_al_2025. All other relevant data and details of resources can be found within the article and its supplementary information.

## Peer review history
The peer review history is available online at https://journals.biologists.com/dev/lookup/doi/10.1242/dev.204811.reviewer-comments.pdf

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
