## [Peer Review File · Development (Cambridge, England)]

A conserved differentiation programme facilitates inhibitory neuron production in the developing mouse and human cerebellum

Jens Bager Christensen, Alex P. A. Donovan, Marzieh Moradi, Giada Vanacore, Mohab Helmy, Adam J. Reid, Jimmy Tsz Hang Lee, Omer Ali Bayraktar, Andrea H. Brand and N. Sumru Bayin

DOI: 10.1242/dev.204811

Editor: François Guillemot

Review timeline

Original submission:	25 March 2025
Editorial decision:	22 April 2025
First revision received:	21 July 2025
Editorial decision:	16 August 2025
Second revision received:	28 October 2025
Accepted:	15 November 2025

Original submission

First decision letter

MS ID#: dev.204811

MS Title: A conserved differentiation program facilitates inhibitory neuron production in the developing mouse and human cerebellum

Authors: Jens Bager Christensen; Alex P. A. Donovan; Marzieh Moradi; Giada Vanacore; Mohab Helmy; Adam J. Reid; Jimmy Tsz Hang Lee; Omer Ali Bayraktar; Andrea H. Brand; N. Sumru Bayin

Article Type: Research Article

Dear Dr Bayin,

I have now received all the referees' reports on the above manuscript, and have reached a decision. The referees' comments are appended below, or you can access them online: please go to:

As you will see, the referees express great interest in your work, but they also have significant criticisms and recommend a substantial revision of your manuscript before we can consider publication. In particular, one of the referees requests that you use additional markers to identify interneurons in the *in vitro* differentiation experiments. If you are able to revise the manuscript along the lines suggested by the referees, which may involve further experiments, I will be happy receive a revised version of the manuscript. Your revised paper will be re-reviewed by the original referees, and acceptance of your manuscript will depend on your addressing satisfactorily their major concerns. Please also note that Development will normally permit only one round of major revision. If it would be helpful, you are welcome to contact us to discuss your revision in greater detail. Please send us a point-by-point response indicating your plans for addressing the referees' comments, and we will look over this and provide further guidance.

Please attend to all of the reviewers' comments and ensure that you clearly highlight all changes made in the revised manuscript. Please avoid using 'Tracked changes' in Word files as these are lost in PDF conversion. I should be grateful if you would also provide a point-by-point response detailing

how you have dealt with the points raised by the reviewers in the 'Response to Reviewers' box. If you do not agree with any of their criticisms or suggestions please explain clearly why this is so.

Reviewer 1

SUMMARY OF THE ADVANCE MADE IN THIS PAPER AND ITS POTENTIAL SIGNIFICANCE TO THE FIELD

Sumru Bayin and her colleagues analyzed the population of cells derived from Nestin-expressing progenitors (NEPs) at the postnatal stage. She isolates Nestin-CFP+ cells from the cerebellum using FACS and performs scRNA-seq followed by clustering analysis. In addition to previously reported (Pakula et al., 2025) populations such as Hopx+ gliogenic NEPs, Ascl1+ neurogenic NEPs, and granule cell progenitors, the authors identify new clusters. Through further subclustering, they provide a detailed analysis of cell populations and infer their lineage relationships.

Based on the scRNA-seq data, the authors conduct immunostaining on mouse brain sections and perform lineage tracing using CreERT2. They find that Ascl1-expressing NEPs give rise to astrocytes or inhibitory neurons, and propose that Foxo1 may act as a regulator of inhibitory neuron differentiation. To test this, they employ an in vitro differentiation system using neurospheres derived from the cerebellum, confirming that Foxo1 is expressed during the differentiation of inhibitory neurons from Ascl1+ NEPs.

To elucidate the relationship between Ascl1 and Foxo1, they overexpress these transcription factors in NEPs via lentiviral transduction and observe neuronal induction. They also show that Ascl1 can induce Foxo1 expression. Knockdown of Foxo1 using shRNAs in the in vitro differentiation system suppresses neuronal differentiation.

Furthermore, combining the in vitro differentiation system with Targeted DamID, the authors identify the target genes of ASCL1 and FOXO1, suggesting that these transcription factors cooperatively regulate gene expression for neuronal differentiation. The overexpression experiments also indicate that Ascl1 promotes proliferation, whereas Foxo1 is involved in cell survival.

Additionally, the scRNA-seq data suggest that Wnt signaling may serve as an upstream regulator of Foxo1 expression. Activation of Wnt signaling with CHIR in vitro promotes nuclear translocation of FOXO1 and enhances neuronal differentiation (albeit less than 2-fold). The authors further demonstrate that human NEPs exhibit similar expression of ASCL1 and FOXO1 under comparable in vitro conditions. Based on these findings, they propose that the ASCL1-WNT-FOXO1 axis plays a crucial role in the generation of inhibitory neurons from NEPs.

SUGGESTIONS TO AUTHORS

The scRNA-seq analysis incorporates cluster analysis, trajectory inference, and in silico overexpression/knockout approaches, employing state-of-the-art methodologies to support the proposed ASCL1-WNT-FOXO1 model. However, the majority of the functional validation is conducted in vitro, making it difficult to determine whether the described processes occur under physiological conditions in vivo. At a minimum, in vivo knockout—or at least knockdown—of Foxo1 would be essential to substantiate its role. Likewise, the identification of ASCL1 and FOXO1 target genes would benefit from ChIP or comparable evidence for selected examples. While the Wnt signaling findings are intriguing, the observed effects are relatively modest (less than twofold) and entirely derived from in vitro systems, which raises concerns regarding their in vivo relevance. Overall, although the study contributes valuable insights into postnatal cerebellar cell differentiation, the findings may be interpreted as specific to in vitro conditions rather than reflecting endogenous developmental processes.

Reviewer 2

SUMMARY OF THE ADVANCE MADE IN THIS PAPER AND ITS POTENTIAL SIGNIFICANCE TO THE FIELD

The control of development of cerebellar inhibitory neurons remains poorly understood, compared to some other regions of the brain. The manuscript by Christensen et al focusses on the transcriptional control of differentiation of cerebellar inhibitory neurons from their progenitors in the early postnatal mouse brain. The study also illustrates parallels between these regulatory mechanisms between the mouse and human brain. Using earlier transcriptomic data (scRNAseq), the authors identify Foxo1 as a potential regulator of neuronal differentiation downstream of a proneural gene Ascl1. They use in vitro neurosphere/adherence culture method to demonstrate functions of Foxo1 and Ascl1 in the differentiation of postmitotic cerebellar inhibitory neurons. Using the in vitro cultures, the authors identify genomic targets of Foxo1 and Ascl1, uncovering both shared and unique genes potentially regulated by these TFs. The results also suggest Foxo1 as a direct target of Ascl1. Ascl1 and Foxo1 were found to have distinct effects on cell proliferation and survival when over-expressed in the differentiating cerebellar neuros cultured in vitro. With a computational analysis of the scRNAseq data, the authors identify Wnt signaling potentially important for the early cerebellar neuron differentiation. They find Wnt pathway activation to increase numbers of Foxo1 expressing cells and postmitotic neurons in the in vitro cultures. Finally, the authors find Ascl1 and Foxo1 also expressed in human cerebellar neuron progenitors and show that activation of Wnt signaling increases numbers of Foxo1 expressing cells and post-mitotic neurons in cultures of human cerebellar neuron progenitors. The results are of high quality, well controlled & analysed, and clearly presented. The study convincingly identifies novel aspects of the differentiation process of cerebellar inhibitory neurons.

SUGGESTIONS TO AUTHORS

I have some concerns related to the analyses of the genomic targets of Foxo1 and Ascl1

1. More detailed description of the DamID experiments and the results are needed (Fig 4B). Statistics of the DamID peaks - number, length? Were the Ascl1 and Foxo1 motifs also enriched in Dam-alone experiments? Were the Foxo1 motifs enriched in Ascl1-Dam? Were the Ascl1 motifs enriched in Foxo1-Dam? What was the overlap between the Foxo1 and Ascl1 peaks?
2. What is the rationale for restricting analyses of target genes to genes with their TSS within +/- 1 kb from Ascl1 or Foxo1 binding site? Most of the true targets will be left out, hampering conclusions.
3. Description of Fig 4D differs in the main text (results 347-349) and the figure legend (1038-1040). Does the Venn diagram show the peaks or associated genes?
4. Overlap between genes/peaks associated with Ascl1 and Foxo1 binding. The nature of this overlap should be better described and discussed. Do the motifs coincide? Are there other features that correlate with the potential Ascl1&Foxo1 co-binding?

Other points

1. Reference to Pakula et al missing
2. Fig. 5A seems same as Fig 1I in terms of trajectories, but the pseudotime is from a different method. Use of different pseudotime should be more clearly explained
3. Methods - Targeted DamID: "All peaks where distancetoFeature was > 1 kb were discarded"?
4. Methods - Clustering: terminology: "Seurat object" instead of "rds object"
5. Methods - Clustering: give a more detailed justification for the clustering resolution
6. Methods-scVelo: was this a steady-state or dynamic model? Was the root selected and if so, how?
7. Methods - CellOracle: Explain the choice of the root cell
8. Methods - Cell2Fate: Explain the data used for training, validation and final analysis
9. 180-181 -sentence difficult to read
10. Fig S1J. In the pseudotime ordered cells, Foxo1 expression occurs before Ascl1? This is contrary to the mode presented in the paper.

Reviewer 3

SUMMARY OF THE ADVANCE MADE IN THIS PAPER AND ITS POTENTIAL SIGNIFICANCE TO THE FIELD

Using single-cell RNA sequencing (scRNA-seq), Christensen et al, identify genetic programs crucial for NEP-to-inhibitory neuron transition. By leveraging a cell culture system, they demonstrate that the Foxo1 transcription factor downstream of Ascl1 enriches the neurogenic identity by defying cell

death. The authors provide evidence that the binding of *Ascl1* to the *Foxo1* promoter marks the beginning of a temporally dynamic regulatory program whose components function independently of each other: while *Ascl1* supports proliferation and neurogenesis, specified neurons rely on *Foxo1* to survive. As for cell-extrinsic factors that trigger *Foxo1* expression, the authors identified a role for Wnt signaling in regulating the *Ascl1*-to-*Foxo1* transition. Remarkably, the *Ascl1*-Wnt-*Foxo1* axis is conserved from mouse to human, albeit protracted in human.

SUGGESTIONS TO AUTHORS

Comments and suggestions:

1. The authors should clearly define "NEP cells" throughout the manuscript. The scRNA-seq dataset from FACS-isolated *Nes*⁺ cells includes various cerebellar cell types. If this is not due to contamination, it raises the possibility that NEPs possess broader developmental potential.
2. The methods state that primary NEPs were isolated from *Nes*-CFP transgenic mice, but do not clarify whether FACS was used to enrich for these cells. The criteria for defining human NEPs are also unclear and should be specified.
3. The central white matter also contains *Nestin*⁺ progenitors, yet this population is not discussed in the study. The authors should clarify whether and how this region was examined.
4. The rationale for excluding certain cell clusters from iterative clustering and trajectory analyses should be clarified. For instance, why were N3 (*Pvalb*⁺) cells excluded? Are these cells considered contaminants, or were they excluded due to being outside the study's focus? Additionally, performing scVelo and Cell2Fate analyses on the full set of *Nes*⁺ cells could provide a more comprehensive view of lineage relationships. Such an inclusive analysis might reveal whether *Ascl1*⁺ NEPs can give rise to *Hopx*⁺ NEPs, or vice versa, helping to resolve potential lineage transitions within the NEP population.
5. The presence of CFP⁺ granule cell progenitors (clusters 2, 5, and 8; *Atoh1*⁺/*Barhl1*⁺) should be explained. Do NEPs normally contribute to the external granule layer?
6. The relationship between the newly defined N1-N10 clusters and the original clusters (0, 1, 3, 4, 6) should be clearly described.
7. The study lacks sufficient evidence supporting interneuron identity. Markers such as *TuJ1* and *Gad1* are not specific. Immunostaining for *Pax2* or other interneuron markers is recommended, especially in the in vitro differentiation assays.
8. The claim that *Ascl1* precedes *Foxo1* during NEP-to-interneuron differentiation is not fully supported by the in vivo data. In Fig. 1H, *Ascl1* appears more broadly expressed than *Foxo1*.
9. In Figs. 2G-J, *Ascl1* and *Foxo1* are not co-expressed in vitro, in contrast to in vivo data. This discrepancy should be addressed.
10. For Figs. 4L-M, the authors should rule out the possibility that EGFP overexpression may induce cell death. If this is the case, it would compromise the validity of EGFP as a negative control and potentially confound the interpretation of the effects observed with *ASCL1* and *FOXO1* overexpression.
11. Multiple testing correction should be applied to the p-values in the motif enrichment analysis (Fig. 4I).

Minor Comments

1. In Fig. 5A, the arrows from *Hopx* to W3 should be discussed. Do the results suggest that *Hopx*⁺ cells also give rise to W3 cells?
2. In Figs. 5C-H, please provide brief explanations of the terms "On" and "Induction state" for readers unfamiliar with the Cell2Fate methodology.
3. In Figs. 5C-G, Module 2 aligns with W4, while Module 4 aligns with W3, where *Foxo1* is expressed. This relationship should be described more clearly in the text.
4. Consider replacing cluster numbers with cell-type annotations where possible to improve accessibility for the reader.
5. In Fig. S1B, organizing module trends into 2-3 general trajectories would enhance readability. The current presentation is difficult to interpret due to overlapping lines and color complexity. Additionally, showing module trends for both P1 and P2 replicates would strengthen the reproducibility of the findings.
6. In line 172, the text refers to gliogenic and neurogenic NEPs undergoing additional subclustering, yet *Hopx*⁺ NEPs were excluded from this analysis. Please clarify this distinction in the text (line 172) and in Figs. 1G, 1I, 1K, 1L, and S1H-I.

7. The heatmap in Fig. S1J lacks an x-axis label. Please add the appropriate labeling for clarity.
8. In Table S7, "Tuckey" should be corrected to "Tukey."
9. The conclusion that Foxo1 inhibition prevents neuron production may be overstated. According to Table S7, Foxo1 shRNA1 does not significantly reduce TUJ1 expression compared to the scramble control. This point should be addressed or qualified.
10. It is unexpected that the number of Ascl1+ cells remains unchanged following CHIR treatment in both mouse and human datasets, given the proposed model in which WNT signaling promotes the transition from an Ascl1+ to a Foxo1+ state. To support the model, it would be helpful to report Ascl1 protein levels and assess their correspondence with Foxo1 expression in the same cells.

First revision

Author response to reviewers' comments

We would like to thank the Reviewers for their constructive comments. We are glad that all reviewers found our study well-designed and exciting. We have addressed their comments by adding new data and editing the text. Importantly, we have made the following changes and additions to our manuscript:

- 1) We included further characterisation of the primary mouse and human NEP neurosphere cultures via scRNA-seq, showing their transcriptomic resemblance to NEPs that are freshly isolated from mouse brain (our dataset) or human fetal brain (Sepp et al., 2024) (Figures S2, S7, Tables S2, S7).
- 2) We assessed cerebellar inhibitory neuron-specific differentiation marker PAX2 during differentiation. (Figure 2G-L, 6D)
- 3) We validated targets identified by ASCL1 and FOXO1 targeted DamID on neonatal mouse brain sections (where *Ascl1*-NEPs and their progeny were labelled using genetic inducible fate mapping) using *in situ* HCR analysis (Figure S5).
- 4) We provided further in-depth analysis of the targeted DamID datasets, including detailed statistics of peaks and their distributions across different genomic elements (Figure S4). In addition, we performed motif enrichment analysis on Dam-alone samples and additional GO-term enrichment analysis on peaks that are within ± 2 kb of transcriptional start sites (Figures 4B and S4, Table S4, S5).
- 5) We included quantification of ASCL1 and FOXO1 double positive cells during differentiation (Figure 2T and 6M) and WNT signalling activation (Figures 5N, 7F), and included a new paragraph in the discussion on the discrepancies between the *Ascl1* mRNA and protein levels.
- 6) We amended our methods as requested by the reviewers and expanded the discussion on the limitations of our study.

Our point-by-point response to the reviewer's comments is below.

Reviewer 1:

SUGGESTIONS TO AUTHORS

The scRNA-seq analysis incorporates cluster analysis, trajectory inference, and *in silico* overexpression/knockout approaches, employing state-of-the-art methodologies to support the proposed ASCL1-WNT-FOXO1 model. However, the majority of the functional validation is

conducted *in vitro*, making it difficult to determine whether the described processes occur under physiological conditions *in vivo*. At a minimum, *in vivo* knockout—or at least knockdown—of *Foxo1* would be essential to substantiate its role. Likewise, the identification of *ASCL1* and *FOXO1* target genes would benefit from ChIP or comparable evidence for selected examples. While the Wnt signaling findings are intriguing, the observed effects are relatively modest (less than twofold) and entirely derived from *in vitro* systems, which raises concerns regarding their *in vivo* relevance. Overall, although the study contributes valuable insights into postnatal cerebellar cell differentiation, the findings may be interpreted as specific to *in vitro* conditions rather than reflecting endogenous developmental processes.

We would like to thank the reviewer for highlighting the key findings of our study. While we agree that validating some of our findings *in vivo* would have been exciting, this is beyond the scope of this manuscript. Importantly, we do not have immediate access to a *Foxo1-flox* allele or equipment to carry out such experiments within the given time frame for these revisions. On the other hand, we do not think that the lack of this analysis undermines our findings. First of all, we would like to highlight that the initial scRNA-seq analysis was performed on freshly isolated cells from the neonatal mouse cerebellum. Furthermore, we confirmed the temporal relationship between *ASCL1* and *FOXO1* using fate mapping in neonatal mice. Finally, we discussed previous research in the field on WNT signalling that is in support of our findings in our discussion.

A major challenge in studying rare cell populations *in vivo* is the limited material for molecular studies. This applies to NEP subpopulations in the neonatal cerebellum as well. This has been impeding the progress in our understanding of the gene regulatory networks that govern the differentiation process in diverse neuron types in brain subregions. Importantly, ChIP and similar approaches have been particularly challenging when dealing with the neonatal cerebellum, where at a given time, we anticipate only <10,000 cells in the populations of interest, and even less for the cell states mentioned here. Furthermore, non-synchronised differentiation states observed at a given snapshot *in vivo* complicate bulk analyses. Therefore, one of the main goals of our study is to demonstrate the utility of primary mouse and human cerebellar progenitor cultures. Using these, we could expand the cells of interest and perform temporally controlled experiments to dissect differentiation processes.

With these constraints in mind, we further validated our results orthogonally *in vivo*. We performed RNA *in situ* HCR analysis for some of the *ASCL1*-Dam and *FOXO1*-Dam targets on neonatal mouse cerebellar sections (Figure S5). In addition, we included new scRNA-seq analysis of our mouse and human primary NEP neurosphere cultures. Integration of these data to the mouse and human NEPs using datasets generated from cells freshly isolated from tissue (mouse dataset used in the manuscript (Figure S2) and previously published human fetal cerebellum snRNA-seq data (Sepp et al., 2024) (Figure S7)) highlighted the transcriptional similarities of the primary cultures to their *in vivo* counterparts. Finally, we assessed expression of *PAX2* during *in vitro* differentiation to further demonstrate the cerebellar inhibitory neuron identity (mouse NEPs: Figure 2G-L, human NEPs: Figure 6D). Additionally, in the revised manuscript, we expanded the section on the caveats of our study to highlight the need for future *in vivo* loss of function studies.

Reviewer 2:

SUGGESTIONS TO AUTHORS

1. More detailed description of the DamID experiments and the results are needed (Fig 4B). Statistics of the DamID peaks - number, length?

We included these statistics of targeted DamID peaks in the revised manuscript (Figure S4).

Were the *Ascl1* and *Foxo1* motifs also enriched in Dam-alone experiments? Were the *Foxo1* motifs enriched in *Ascl1*-Dam? Were the *Ascl1* motifs enriched in *Foxo1*-Dam?

We performed motif enrichment analysis in Dam-alone and observed significant enrichment of the *ASCL1* motif at both day 2 and 6, while the *FOXO1* motif was significantly enriched only at

day 6. We also observed that the ASCL1 motif was significantly enriched in FOXO1- Dam peaks at both time points and vice versa. This aligns with previous reports from other model systems (Park et al., 2017). The adjusted p-values for ASCL1 and FOXO1 motifs in all conditions are presented in **Figure 4B**, and the complete list of enriched motifs in ASCL1- Dam, FOXO-Dam and Dam-alone can be found in **Table S4**.

What was the overlap between the Foxo1 and Ascl1 peaks?

We observed 29.81% and 8.41% overlap between the reproducible significant peaks from FOXO1-Dam and ASCL1-Dam occurring ± 1 kb of a TSS at day 2 and day 6, respectively. The overlap for peaks within 2kb of a TSS was 31.20% at day 2 and 8.69% at day 6 of differentiation. Finally, when all peaks were analysed, the overlap between peaks was 37.85% at day 2 and 13.57% at day 6. These statistics are reported in the revised manuscript (pg. 14, lines 423-427).

2. What is the rationale for restricting analyses of target genes to genes with their TSS within +/- 1 kb from Ascl1 or Foxo1 binding site? Most of the true targets will be left out, hampering conclusions.

We repeated the analysis also with the peaks within 2kb of a TSS. While we observed an increase in the number of genes associated with peaks, we did not observe any significant changes in the GO terms identified that could suggest additional functions. These results are now presented in **Figure S4** and **Table S5**. While distal binding ($> |2\text{kb}|$) from the TSS could be very interesting, it would be hard to relate it to gene expression without exploring 3D chromatin conformation and topologically associated domains. Such analysis is beyond the scope of this project.

3. Description of Fig 4D differs in the main text (results 347-349) and the figure legend (1038-1040). Does the Venn diagram show the peaks or associated genes?

We apologise for the confusion. The Venn diagrams in **Figures 4E-F** and **S4D-E** refer to the peaks associated with genes. We clarified this in the revised manuscript and the respective figure legends.

4. Overlap between genes/peaks associated with Ascl1 and Foxo1 binding. The nature of this overlap should be better described and discussed. Do the motifs coincide? Are there other features that correlate with the potential Ascl1&Foxo1 co-binding?

We included the percentages of the overlap between the genes/peaks associated with both ASCL1-Dam and FOXO1-Dam binding in the revised manuscript and included motif enrichment analysis on the Dam-alone peaks of the genes that are bound by both ASCL1- Dam and FOXO1-Dam (**Table S4**). Given the lower number of peaks used for this analysis, fewer significant motifs (< 15) were identified with higher-than-usual adj p-values (Table S4). Some of the consistently significant motifs detected in ASCL1-Dam, FOXO1-Dam and Dam- alone include other bHLH neurogenic genes, SOX family genes, TCF4 and AP1, suggesting their potential roles during differentiation. We highlighted these findings in the results section (page 13, lines 403-407, page 14, lines 427-432).

Due to the technical aspects of the targeted DamID library preparation, which selects for the methylated-GATC motifs around the potential binding sites, multiple motifs for a transcription factor are detected within the fragments, and the data lacks the resolution to annotate which ones are bound by the transcription factors. Therefore, we cannot confidently comment on whether the motifs coincide or not. We expanded the discussion on the limitations of targeted DamID data (page 24, lines 754-755).

Minor comments

1. Reference to Pakula et al missing.

We updated the reference.

2. Fig. 5A seems same as Fig 1I in terms of trajectories, but the pseudotime is from a different method. Use of different pseudotime should be more clearly explained

The trajectories shown in **Figure 5A-B** were generated with Cell2Fate (as opposed to scVelo, **Figure 1I**). Although similar, these trajectories are not the same. The Cell2Fate was able to provide clearer trajectories within similar cells/cellular states due to the inference method that is applied in the velocity prediction (Aivazidis et al., 2025), whereas scVelo predicted the bifurcation of the bipotent cells better, due to the inherent differences in both algorithms (Bergen et al., 2020; La Manno et al., 2018). A systematic benchmarking between the two algorithms is lacking. Given our prior knowledge on the WM NEPs and their progeny, we took advantage of both to dissect the earlier and later events during NEP-to- inhibitory neuron differentiation. We included an explanation for the different choice of Cell2Fate vs. scVelo where relevant (page 16, lines 504-510).

3. Methods - Targeted DamID: "All peaks where distancetoFeature was > 1 kb were discarded"?

We rephrased the sentence.

4. Methods - Clustering: terminology: "Seurat object" instead of "rds object"

We changed the terminology as requested.

5. Methods - Clustering: give a more detailed justification for the clustering resolution

We expanded the justification for the clustering resolution in the *Methods*.

6. Methods-scVelo: was this a steady-state or dynamic model? Was the root selected and if so, how?

We used a steady-state model due to the shape of the phase plots as suggested by the developers. The analysis was unsupervised, and no root cell was chosen. We included these details in the *Methods*.

7. Methods - CellOracle: Explain the choice of the root cell

The details of the root cell choice have been added to the *Methods* (page 37, lines 1193-1196).

8. Methods - Cell2Fate: Explain the data used for training, validation and final analysis

The information regarding what data was used to train the model has been added to the *Methods* (page 38, lines 1205-1210). The training was validated based on the convergence of the evidence lower bound (ELBO) (Aivazidis et al., 2025).

9. 180-181 -sentence difficult to read

We edited the sentence.

10. Fig S1J. In the pseudotime-ordered cells, Foxo1 expression occurs before Ascl1? This is contrary to the mode presented in the paper.

The heatmap in (old) Figure S1J showed the normalised expression, which looks misleading, as pointed out by Reviewers 2 and 3. We changed this graph to better reflect expression levels of *Ascl1* and *Foxo1* along the pseudotime (**Figure S1K**). As previously discussed, while *Ascl1* expression spanned a broader pseudotime period and fluctuated, it preceded *Foxo1* mRNA levels along the differentiation trajectory. This was further supported by our histological analysis on cerebellar sections (**Figure 1L-M**) and during *in vitro* differentiation (**Figure 2M-R**). However, interestingly, our observations suggest that the *Ascl1* mRNA and protein levels may differ during different stages of differentiation. We stated these observations in the revised manuscript and included a new paragraph in the discussion to elaborate on potential mechanisms (page 22, lines 679-694).

Reviewer 3:**SUGGESTIONS TO AUTHORS**

1. The authors should clearly define "NEP cells" throughout the manuscript. The scRNA-seq dataset from FACS-isolated Nes⁺ cells includes various cerebellar cell types. If this is not due to contamination, it raises the possibility that NEPs possess broader developmental potential.

We clarified the description of NEPs and what the other Nes⁺ cells in the dataset represent. The Nes-Cfp reporter and the NEP subtypes were characterised in previous work (Bayin et al., 2021; Cerrato et al., 2018). We elaborated further on these details to help the reader and highlighted the relevant cell types for our study. We hope that this clarifies any confusion around the populations of interest and broader expression of Nes-CFP transgene in the cerebellum (page 6, lines 153-168, page 7, lines 181-184, 190-192, 196-201).

2. The methods state that primary NEPs were isolated from Nes-CFP transgenic mice, but do not clarify whether FACS was used to enrich for these cells. The criteria for defining human NEPs are also unclear and should be specified.

We apologise for the lack of details. We explained how the primary cultures were generated in the Methods in detail. hNEPs were defined as the NES/SOX2 expressing cells of the fetal cerebellum, which encompass the progenitors that were previously annotated as inhibitory neuron progenitors and Bergmann glial progenitors. The primary NEP cultures are defined by their ability to form neurospheres and endogenous expression of Sox2, Nes, and other NEP subtype markers, and the lack of rhombic lip markers (page 25, lines 793-805). The addition of new scRNA-seq on the mouse and human primary cultures further supports our findings regarding the *in vivo* relevance of our cultures (Figures S2 and S7). Interestingly, primary hNEP neurosphere cultures showed greater cellular diversity compared to the primary mouse NEP neurosphere cultures. Our future work will focus on dissecting the cellular heterogeneity of hNEPs and their relevance to human cerebellum development.

3. The central white matter also contains Nestin⁺ progenitors, yet this population is not discussed in the study. The authors should clarify whether and how this region was examined.

The scRNA-seq data was generated via FACS-isolating Nes-CFP⁺ cells from neonatal mouse cerebella. Our previous analysis also shows that there are rare CFP⁺ cells in the deep white matter, but these cells represent <10% of all the CFP⁺ cells in the cerebellum and therefore are likely underrepresented in our dataset (as reported in Bayin et al., 2021, Figure S2I-K). Finally, these cells do not contribute to ML inhibitory neuron production but rather generate glia in the deep white matter after birth (previously reported in Bayin et al., 2021, Figure S6). Therefore, these cells are out of the scope of this manuscript.

4. The rationale for excluding certain cell clusters from iterative clustering and trajectory analyses should be clarified. For instance, why were N3 (Pvalb⁺) cells excluded? Are these cells considered contaminants, or were they excluded due to being outside the study's focus?

We expanded the narrative on our logic for excluding some clusters in the Methods and throughout the text. Briefly, this is based on the previous fate mapping work of ours and others, and the clusters are omitted due to their irrelevance to the ML inhibitory neuron lineage. Specifically, analysis of Pvalb⁺ cells in the neonatal cerebellum via RNA in situ hybridisation (Figure S1H) shows that at the relevant age (P4), Pvalb is not expressed in lobule WM (where the neurogenic Ascl1-NEPs) but is only expressed in the deep white matter. This also highlights the potential characteristics of some the deep white matter Nes⁺ cells mentioned in the previous comment. However, given that they are not related to the ML inhibitory neurons and represent a different inhibitory neuron population that is born between embryonic day 10-15, and are not generated

from lobule *Ascl1*-NEPs, we omitted this cluster from further analysis (page 7, lines 182-184).

Additionally, performing scVelo and Cell2Fate analyses on the full set of *Nes*⁺ cells could provide a more comprehensive view of lineage relationships. Such an inclusive analysis might reveal whether *Ascl1*⁺ NEPs can give rise to *Hopx*⁺ NEPs, or vice versa, helping to resolve potential lineage transitions within the NEP population.

The lineage propensities of NEP subtypes with distinct localisation within the cerebellum cytoarchitecture are well-defined by previous fate mapping and clonal analysis (Bayin et al., 2021; Cerrato et al., 2018; Joyner and Bayin, 2022). Therefore, not all *Nes*⁺ cells in the scRNA-seq have a direct hierarchical relationship with each other. On the other hand, the gene regulatory networks that drive the differentiation processes remain unknown. Therefore, in this manuscript, we wanted to highlight the power of iterative subclustering to isolate related lineages for downstream analyses. To aid this understanding, we further clarified the subclustering logic and the rationale for omitting different clusters for downstream analysis.

5. The presence of CFP⁺ granule cell progenitors (clusters 2, 5, and 8; *Atoh1*⁺/*Barhl1*⁺) should be explained. Do NEPs normally contribute to the external granule layer?

We included an explanation for the CFP⁺ granule cell progenitors in the manuscript. These cells have been previously reported; however, whether they are rhombic lip- or ventricular zone-derived remains unclear. Under physiological conditions, gliogenic and neurogenic NEPs (*Hopx*⁺ and *Ascl1*⁺) do not give rise to granule cell progenitors (page 6, lines 153-168). However, upon injury or ablation of granule cell progenitors at birth, specifically the NEPs in the Bergmann glia layer, are able to undergo adaptive reprogramming and replenish the lost cells (Bayin et al., 2021; Wojcinski et al., 2017). While this injury-induced fate switch is not relevant to this manuscript, it highlights the importance of understanding how different GRNs regulate fate decisions and differentiation during development, in order to parse out the injury-induced regenerative responses.

6. The relationship between the newly defined N1-N10 clusters and the original clusters (0, 1, 3, 4, 6) should be clearly described.

We improved the explanation of our data set and the populations detected in the initial data. We have also clarified the subclustering logic to help the reader and elaborated further on other NEP subpopulations and their lineages. A Sankey plot demonstrating the relationship between clusters from different subsets has been added to increase clarity (Figure S1G). The lineage relationships of NEP subtypes are also summarised in Figure 1C.

7. The study lacks sufficient evidence supporting interneuron identity. Markers such as *TuJ1* and *Gad1* are not specific. Immunostaining for *Pax2* or other interneuron markers is recommended, especially in the in vitro differentiation assays.

We have incorporated additional immunostaining for *PAX2* (and *LHX5* in human) on our differentiated mouse and human NEP cultures to characterise interneuron identity (Figures 2G-L and 6E-D).

8. The claim that *Ascl1* precedes *Foxo1* during NEP-to-interneuron differentiation is not fully supported by the in vivo data. In Fig. 1H, *Ascl1* appears more broadly expressed than *Foxo1*.

We have elaborated on the dynamics of *Ascl1* mRNA and protein expression throughout the *Ascl1*-NEP to inhibitory neuron differentiation in the text and improved the manuscript to highlight these differences better (Figure S1K) (page 22, lines 679-694).

9. In Figs. 2G-J, *Ascl1* and *Foxo1* are not co-expressed in vitro, in contrast to in vivo data. This discrepancy should be addressed.

We included new images and quantification of the percentages of the *ASCL1* and *FOXO1* double

positive cells during *in vitro* differentiation, highlighting the timepoint when the highest percentages were observed, and expanded the discussion on the ASCL1⁺/FOXO1⁺ state in the manuscript (**Figures 2M-Q and T**) (page 22, lines 679-694).

10. For Figs. 4L-M, the authors should rule out the possibility that EGFP overexpression may induce cell death. If this is the case, it would compromise the validity of EGFP as a negative control and potentially confound the interpretation of the effects observed with ASCL1 and FOXO1 overexpression.

We observed increased cell death with EGFP OE compared to the parent rTTA NEPs, as suspected by the reviewer (**Rebuttal Figure 1**). However, we respectfully disagree with the comment that EGFP is not the appropriate control for such studies. To the contrary, it is important to have the EGFP OE as a control to provide a baseline for all overexpression studies, given the cellular stress the cells endure due to ectopic overexpression of large amounts of proteins.

Rebuttal Fig. 1. Density of TUNEL particles in different overexpression conditions (one-way ANOVA, $F(3, 12) = 7.362$, $p=0.0047$, $n=4$). Graphs are mean \pm s.e.m. Significant p-values for Tukey's multiple comparisons test are shown.

11. Multiple testing correction should be applied to the p-values in the motif enrichment analysis (Fig. 4I).

Thank you for highlighting this. We performed Bonferroni multiple testing correction to the motif enrichment analysis and reported them in **Figure 4B** and **Table S4**.

Minor Comments

1. In Fig. 5A, the arrows from Hopx to W3 should be discussed. Do the results suggest that Hopx+ cells also give rise to W3 cells?

The Cell2Fate analysis identified astrocytes (W6) as an early population along with WM- NEPs (W4) in inferred time (**Figure 5A-B**), in contrast to the scVelo, which detected only W4 as the top of the hierarchy, while not resolving the lineage progression along the committed neurons as convincingly (**Figure 1I**). Whether these immature astrocytes in W6 are fully committed or still maintain bipotent characteristics/neurogenic potential requires further analysis. We included a sentence about this in the revised manuscript (page 17, lines 507-510). The discrepancy between the two different pseudotemporal analyses highlights why we utilised both for different questions (Please see answer to Reviewer 2 Minor comment #2) and the importance of validation.

2. In Figs. 5C-H, please provide brief explanations of the terms "On" and "Induction state" for readers unfamiliar with the Cell2Fate methodology.

We included brief explanations as requested (page 17, lines 515-518).

3. In Figs. 5C-G, Module 2 aligns with W4, while Module 4 aligns with W3, where Foxo1 is expressed. This relationship should be described more clearly in the text.

We highlighted these correlations in the text (page 17, lines 519-523).

4. Consider replacing cluster numbers with cell-type annotations where possible to improve accessibility for the reader.

We tried to implement this where possible. However, in some cases, where the clusters reflect cellular states rather than cell types, we avoided cell-type annotations.

5. In Fig. S1B, organizing module trends into 2-3 general trajectories would enhance readability. The current presentation is difficult to interpret due to overlapping lines and color complexity. Additionally, showing module trends for both P1 and P2 replicates would strengthen the reproducibility of the findings.

We changed how we presented the data for clarity and included information on individual replicates (P1 and P2) where possible (Figure 1SB, E and J).

6. In line 172, the text refers to gliogenic and neurogenic NEPs undergoing additional subclustering, yet *Hopx*⁺ NEPs were excluded from this analysis. Please clarify this distinction in the text (line 172) and in Figs. 1G, 1I, 1K, 1L, and S1H-I.

This subclustering aimed at focusing on WM NEP populations, where the omitted *Hopx*⁺ NEPs represent the Bg progenitors based on their *Gdf10* expression (Figure 1F). The *Hopx*⁺ NEPs residing in the Bergmann glia layer and the WM give rise to distinct lineages as previously reported by clonal analysis (Cerrato et al., 2018). We modified the text to better explain the subclustering logic where possible.

7. The heatmap in Fig. S1J lacks an x-axis label. Please add the appropriate labeling for clarity.

We replaced this figure to better represent the relationship between *Ascl1* and *Foxo1* mRNA levels across pseudotime (Figure S1K).

8. In Table S7, "Tuckey" should be corrected to "Tukey."

We apologise for the mistake. It has been corrected in the revised manuscript.

9. The conclusion that *Foxo1* inhibition prevents neuron production may be overstated. According to Table S7, *Foxo1* shRNA1 does not significantly reduce TUJ1 expression compared to the scramble control. This point should be addressed or qualified.

We incorporated a new biological replicate into this analysis, which shows a significant reduction in TUJ1 for both shRNAs (Figure 3L-Q). However, we acknowledged that the knockdown, and therefore the downstream effects, are partial. We have highlighted this in the results section (page 12, lines 362-366).

10. It is unexpected that the number of *Ascl1*⁺ cells remains unchanged following CHIR treatment in both mouse and human datasets, given the proposed model in which WNT signaling promotes the transition from an *Ascl1*⁺ to a *Foxo1*⁺ state. To support the model, it would be helpful to report *Ascl1* protein levels and assess their correspondence with *Foxo1* expression in the same cells.

Thank you for highlighting this point. We included the percentage of ASCL1 and FOXO1 double positive cells upon CHIR treatment to address this point (Figure 4N and 4F). Although not significant, we observed a trend towards an increase in both mouse and human primary NEP cultures, in line with our hypothesis that WNT activation promotes the transition to a FOXO1⁺ state. On the other hand, we did not observe significant changes in the fluorescent intensity of ASCL1 upon CHIR treatment (Rebuttal Figure 2A), however, given the rarity of double positive cells, it is hard to draw conclusions with respect to protein levels. When we quantified the levels of nuclear FOXO1 via fluorescence intensity within the ASCL1⁺ cells, we did not observe any

correlation (**Rebuttal Figure 2B**). We believe that more accurate measures are needed to come up with conclusions regarding protein levels. To this point, we expanded the discussion on the dynamic regulation of *Ascl1* mRNA and protein expression in the revised discussion (as this also relates to points raised by Review 2 minor comment #10, and Reviewer 3 main points #8 and 9) (page 22, lines 679-694).

Rebuttal Fig. 2. Changes in ASCL1 intensity upon CHIR treatment (primary mouse NEPs, n=6 experiments). A. ($p=0.92$, Student's t-test) B. (DMSO: $R^2=0.06$, CHIR: $R^2=0.02$)

Rebuttal References:

- Aivazidis, A., Memi, F., Kleshchevnikov, V., Er, S., Clarke, B., Stegle, O. and Bayraktar, O. A. (2025). Cell2fate infers RNA velocity modules to improve cell fate prediction. *Nat. Methods*.
- Bayin, N. S., Mizrak, D., Stephen, D. N., Lao, Z., Sims, P. A. and Joyner, A. L. (2021). Injury-induced ASCL1 expression orchestrates a transitory cell state required for repair of the neonatal cerebellum. *Sci. Adv.* **7**, eabj1598.
- Bergen, V., Lange, M., Peidli, S., Wolf, F. A. and Theis, F. J. (2020). Generalizing RNA velocity to transient cell states through dynamical modeling. *Nat. Biotechnol.* **38**, 1408-1414.
- Cerrato, V., Parmigiani, E., Figueres-Oñate, M., Betizeau, M., Aprato, J., Nanavaty, I., Berchiolla, P., Luzzati, F., de'Sperati, C., López-Mascaraque, L., et al. (2018). Multiple origins and modularity in the spatiotemporal emergence of cerebellar astrocyte heterogeneity. *PLOS Biol.* **16**, e2005513.
- Joyner, A. L. and Bayin, N. S. (2022). Cerebellum lineage allocation, morphogenesis and repair: impact of interplay amongst cells. *Development* **149**, dev185587.
- La Manno, G., Soldatov, R., Zeisel, A., Braun, E., Hochgerner, H., Petukhov, V., Lidschreiber, K., Kastrioti, M. E., Lönnerberg, P., Furlan, A., et al. (2018). RNA velocity of single cells. *Nature* **560**, 494-498.
- Park, N. I., Guilhamon, P., Desai, K., McAdam, R. F., Langille, E., O'Connor, M., Lan, X., Whetstone, H., Coutinho, F. J., Vanner, R. J., et al. (2017). ASCL1 Reorganizes Chromatin to Direct Neuronal Fate and Suppress Tumorigenicity of Glioblastoma Stem Cells. *Cell Stem Cell* **21**, 209-224.e7.
- Sepp, M., Leiss, K., Murat, F., Okonechnikov, K., Joshi, P., Leushkin, E., Spänig, L., Mbengue, N., Schneider, C., Schmidt, J., et al. (2024). Cellular development and evolution of the mammalian cerebellum. *Nature* **625**, 788-796.
- Wojcinski, A., Lawton, A. K., Bayin, N. S., Lao, Z., Stephen, D. N. and Joyner, A. L. (2017). Cerebellar granule cell replenishment postinjury by adaptive reprogramming of Nestin+ progenitors. *Nat. Neurosci.* **20**, 1361-1370.

Second decision letter

MS ID#: dev.204811R1

MS Title: A conserved differentiation program facilitates inhibitory neuron production in the developing mouse and human cerebellum

Authors: Jens Bager Christensen; Alex P. A. Donovan; Marzieh Moradi; Giada Vanacore; Mohab Helmy; Adam J. Reid; Jimmy Tsz Hang Lee; Omer Ali Bayraktar; Andrea H. Brand; N. Sumru Bayin
Article Type: Research Article

Dear Dr Bayin,

I have now received the reports of the three referees who reviewed the first version of your manuscript and I have reached a decision. The referees' comments are appended below, or you can access them online: please go to: *****

As you will see, all the referees remain greatly interested in your work, but they still have significant criticisms and recommend an additional round of substantial revisions of your manuscript before we can consider publication. If you are able to revise the manuscript along the lines suggested, which may involve further experiments, I will be happy to receive a revised version of the manuscript. Your revised paper will be re-reviewed by the original referees and acceptance of your manuscript will depend on your addressing satisfactorily the reviewers' major concerns.

If it would be helpful, you are welcome to contact us to discuss your revision in greater detail. Please send us a point-by-point response indicating your plans for addressing the referees' comments, and we will look over this and provide further guidance.

Please attend to all of the reviewers' comments and ensure that you clearly highlight all changes made in the revised manuscript. Please avoid using 'Tracked changes' in Word files as these are lost in PDF conversion. I should be grateful if you would also provide a point-by-point response detailing how you have dealt with the points raised by the reviewers in the 'Response to Reviewers' box. If you do not agree with any of their criticisms or suggestions please explain clearly why this is so.

Reviewer 1

I believe the authors have not fully addressed my previous comments. While I understand the quantitative challenges of performing *in vivo* analyses, as the authors point out, this difficulty does not justify omitting a clear explanation of the limitations of their *in vitro* model, particularly the neurosphere assay system. As far as I know, cerebellar neurosphere assays have not been as thoroughly validated for their resemblance to *in vivo* conditions compared to telencephalic neurospheres. The analyses presented in this manuscript are also rather limited in this regard. I believe the authors should clearly present these as neurosphere-based *in vitro* experiments, although it is of course acceptable to discuss potential similarities in the Discussion.

In Figure S2, the authors present neurosphere assays using P1 Nes-Cfp and Swiss Bester cerebella and claim that the results are comparable to those from "All Nes-CFP+" cells. However, I think it is an overstatement to conclude that these results reflect normal development solely based on this similarity. At the very least, a comparison should be made with the "All Nes-CFP+" population shown in Figure 1E, ideally selecting cells with similar marker expression (e.g., N or W categories). Figure S2 does not seem to provide such data.

While I acknowledge that the authors used advanced techniques to explore the regulatory hierarchy of transcription factors, I believe that at least some experimental validation of the key results is necessary. Although it may not be feasible to perform *in vivo* knockout of *Ascl1*, shRNA-mediated knockdown could be done in the neurosphere assay system. For example, the authors could examine whether *Ascl1* knockdown affects *Foxo1* expression.

DamID is indeed a powerful method, but in the present study it is applied in the context of neurosphere cultures and lentiviral overexpression. While I appreciate that Figure S5 shows co-expression of ASCL1 targets and FOXO1 via HCR, these data are localized and do not provide a comprehensive evaluation. It would be more convincing if ChIP data for representative target genes were provided, particularly since such experiments should be feasible in the neurosphere culture system.

The gene induction effect of CHIR, a Wnt pathway activator, is reported to be only ~1.6-fold, which is extremely modest. This level of induction typically suggests indirect regulation, and alternative interpretations should be considered.

Minor Comments:

The CellOracle analysis is purely an in silico prediction, and experimental validation is necessary. The description in Lines 216-225 is misleading and should be revised accordingly.

Reviewer 2

SUMMARY OF THE ADVANCE MADE IN THIS PAPER AND ITS POTENTIAL SIGNIFICANCE TO THE FIELD

See my earlier review.

SUGGESTIONS TO AUTHORS

The authors have addressed most of the points I raised in my earlier review.

I still have some concerns:

1. As requested, the authors analysed Ascl1 and Foxo1 motifs in the Dam-alone peaks. They found the motifs enriched and this is now mentioned in the manuscript. However, the authors do not consider or discuss the implications of this finding to their conclusions. As Dam-alone was used as a negative control, are the conclusions on the Ascl1-Dam and Foxo1-Dam still valid? If Dam-alone provides a baseline control, one would expect the Ascl1/Foxo1 motifs to be enriched more in Ascl1/Foxo1-Dam compared to Dam-alone. Although the p-values given (Fig 4B,C) may be consistent with this, no enrichment scores are shown and it is unclear if the analyses support the conclusions. This should be addressed. Dam-alone likely represents open chromatin regions - how this affects the interpretation of the results, should be described more clearly. This includes the conclusions of the target co-regulation by Ascl1-Dam and Foxo1-Dam. Currently, it is not even mentioned how the Dam-alone peaks were defined.
2. The authors could mention as a limitation of the study that the distal regulatory elements are excluded from the target analyses of Ascl1 and Foxo1.
3. In Cell2Fate analyses, it should be explained why the authors decided to focus on modules 2, 4 and 6. Furthermore, they should explain the selection of the GO terms shown in Fig 5F-H. For example, the GO terms of the module 2 (Table S6) include both positive and negative regulation of Wnt signalling. In the Fig 5F and in the text, the authors emphasize positive regulation of Wnt signalling in the module 2.
4. There are a few unclear sentences in the manuscript. For example line 164-165, line 169-170, line 267-268

Reviewer 3

SUMMARY OF THE ADVANCE MADE IN THIS PAPER AND ITS POTENTIAL SIGNIFICANCE TO THE FIELD

SUGGESTIONS TO AUTHORS

In the revised manuscript, the authors have addressed most of our previous concerns, with the exception of one point detailed below. The newly provided data have strengthened the overall quality of the paper.

Given the senior authors' previous publications, the substantial number of GCPs recovered in the dataset is somewhat unexpected. The response to our original Point 5 is not entirely accurate.

While several studies have reported Nes⁺ cells in the EGL, none have demonstrated co-expression of Nes and Atoh1. Notably, Peng et al. (Nature Neuroscience, 2013) showed that Nes⁺ cells in the EGL are negative for Atoh1. The authors should revise the manuscript to accurately describe this Nes⁺/Atoh1⁺ population and clarify its novelty.

The study by Peng et al. (Nature Neuroscience, 16(12):1737-44, 2013) is highly relevant and should be cited.

Second revision

Author response to reviewers' comments

We would like to thank the Reviewers for their constructive comments and continued interest in our manuscript. In this version of our manuscript, we addressed the outstanding minor editorial comments of Reviewers 2 and 3 in the text. Additionally, we included new *in vivo* and *in vitro* evidence supporting the relationship between ASCL1 and FOXO1, as requested by Reviewer 1. Briefly, we have made the following changes and additions to our manuscript:

- 1) We added new data confirming that FOXO1 levels are reduced upon loss of ASCL1 in a conditional knockout mouse model *in vivo* and shRNA-mediated knockdown experiments *in vitro* (Figure S5).
- 2) We included the enrichment values for targeted DamID motif analysis in Figure 4B, C and Table S4, updated the relevant Methods section and elaborated on the biological meaning of the results.
- 3) We amended our methods and expanded the discussion on the limitations of our study. We also significantly edited the manuscript to highlight whether the conclusions were drawn from *in vivo*, *in vitro* or *in silico* analyses.

Our point-by-point response to the reviewer's comments is below.

Reviewer 1:

1) I believe the authors have not fully addressed my previous comments. While I understand the quantitative challenges of performing *in vivo* analyses, as the authors point out, this difficulty does not justify omitting a clear explanation of the limitations of their *in vitro* model, particularly the neurosphere assay system. As far as I know, cerebellar neurosphere assays have not been as thoroughly validated for their resemblance to *in vivo* conditions compared to telencephalic neurospheres. The analyses presented in this manuscript are also rather limited in this regard. I believe the authors should clearly present these as neurosphere-based *in vitro* experiments, although it is of course acceptable to discuss potential similarities in the Discussion.

Thank you for your comments. In the previous revision, we included detailed characterisation of the primary neurosphere cultures and their transcriptional similarities to their *in vivo* counterparts. In this round of revision, we further discussed the limitations of our *in vitro* models in the discussion. Although with limitations, we believe that the methodology demonstrated in our manuscript will be a useful tool for the community.

2) In Figure S2, the authors present neurosphere assays using P1 Nes-Cfp and Swiss Bester cerebella and claim that the results are comparable to those from "All Nes-CFP+" cells. However, I think it is an overstatement to conclude that these results reflect normal development solely based on this similarity. At the very least, a comparison should be made with the "All Nes-CFP+" population shown in Figure 1E, ideally selecting cells with similar marker expression (e.g., N or W categories). Figure S2 does not seem to provide such data.

Analysis of transcriptional resemblance within all Nes-CFP+ cells shows that the neurosphere cultures maintain the neurogenic and gliogenic NEP identity, and do not have transcriptional similarity to GCPs or other parenchymal cells like microglia that are present in the all NEP-CFP+ cell

data set (Figure 1E). On the other hand, mapping directly to all NEP (N) or white matter NEP (W) subsets would not show whether there are other non-NEP-like cells within our cultures, but rather highlight which NEP subtypes/states would likely be present in the primary neurosphere cultures. As suggested, we included the requested analyses in Figure S2. Mapping to all categories further demonstrates the transcriptional relevance of the primary cultures to the respective NEP subtypes and/or states.

3) While I acknowledge that the authors used advanced techniques to explore the regulatory hierarchy of transcription factors, I believe that at least some experimental validation of the key results is necessary. Although it may not be feasible to perform *in vivo* knockout of *Ascl1*, shRNA-mediated knockdown could be done in the neurosphere assay system. For example, the authors could examine whether *Ascl1* knockdown affects *Foxo1* expression.

We have performed *in vitro* and *in vivo* loss-of-function analysis for ASCL1 and included these new results in Figure S5. Briefly, we utilised lentiviral based shRNA-mediated knockdown for ASCL1 in primary mouse NEP cultures and showed that there is a partial but significant reduction in the nuclear FOXO1 levels during differentiation at day 4 upon ASCL1 knockdown compared to control cells (Figure S5A-C). Importantly, we also recently generated *Ascl1*-NEP conditional knockout animals (*Ascl1^{CreERT2/fl}*). These pups were given Tamoxifen at P0. Histological analysis of ASCL1⁺ and FOXO1⁺ cell numbers at P3 showed a significant reduction in both cell type densities in the WM compared to the *Ascl1^{fl/fl}* control littermates (Figure S5D-H), further confirming that FOXO1 levels are regulated by ASCL1 in the postnatal developing cerebellum.

4) DamID is indeed a powerful method, but in the present study it is applied in the context of neurosphere cultures and lentiviral overexpression. While I appreciate that Figure S5 shows co-expression of ASCL1 targets and FOXO1 via HCR, these data are localized and do not provide a comprehensive evaluation. It would be more convincing if CHIP data for representative target genes were provided, particularly since such experiments should be feasible in the neurosphere culture system.

Thank you for highlighting your concerns regarding the DamID data. We would like to point out that the TF-Dam fusion proteins are expressed downstream of a transcriptional stop signal and can only be produced by rare events of transcriptional read-through. This limits the expression of the fusion proteins to the physiological range and is actually required for the experiments to work due to the toxicity of the dam methylase if it is allowed to accumulate.

As noted by the reviewer, our HCR analysis on neonatal cerebellar sections (Figure S6) clearly demonstrate the overlap of ASCL1 and/or FOXO1 targets with their respective transcription factors within the neurogenic-NEP population. While we appreciate that these analyses were not global, they strongly support our DamID data. Moreover, although the reviewer is now suggesting chromatin pull-down experiments *in vitro*, some of the important constraints we mentioned in our previous rebuttal still apply. In our hands, due to the transient nature of the progenitor populations and their low frequency in our cultures (Figure 2R-S), performing chromatin pull-down experiments in the differentiated cultures has been challenging. This is the main motivation for performing targeted DamID, where the signal accumulates over time (rather than being a static snapshot from a pull-down) and therefore provides identification of weaker/underrepresented targets. We highlighted these challenges and the need to validate these findings using orthogonal methods that measure native protein binding in the Limitations section of our Discussion.

The gene induction effect of CHIR, a Wnt pathway activator, is reported to be only ~1.6-fold, which is extremely modest. This level of induction typically suggests indirect regulation, and alternative interpretations should be considered.

We discussed possible explanations of the modest phenotype in the text.

Minor Comments:

The CellOracle analysis is purely an *in silico* prediction, and experimental validation is necessary. The description in Lines 216-225 is misleading and should be revised accordingly.

We modified the section title and text to reflect that the conclusions are based on *in silico* predictions.

Reviewer 2:

The authors have addressed most of the points I raised in my earlier review. I still have some concerns:

1. As requested, the authors analysed Ascl1 and Foxo1 motifs in the Dam-alone peaks. They found the motifs enriched and this is now mentioned in the manuscript. However, the authors do not consider or discuss the implications of this finding to their conclusions. As Dam-alone was used as a negative control, are the conclusions on the Ascl1-Dam and Foxo1-Dam still valid? If Dam-alone provides a baseline control, one would expect the Ascl1/Foxo1 motifs to be enriched more in Ascl1/Foxo1-Dam compared to Dam-alone. Although the p-values given (Fig 4B,C) may be consistent with this, no enrichment scores are shown and it is unclear if the analyses support the conclusions. This should be addressed. Dam-alone likely represents open chromatin regions - how this affects the interpretation of the results, should be described more clearly. This includes the conclusions of the target co-regulation by Ascl1-Dam and Foxo1-Dam. Currently, it is not even mentioned how the Dam-alone peaks were defined.

Thank you for highlighting this point. We apologise for the missing information. We included the missing information in the Methods and added the enrichment scores for motif analysis to **Figure 4B,C** and **Table S4**, to further highlight the strength of the enrichment in ASCL1-Dam and FOXO1-Dam conditions, in line with the expectations. The biological significance of these findings is also explained in the revised manuscript.

The enrichment of ASCL1 and FOXO1 motifs in Dam-alone condition suggests the permissiveness of the chromatin at these stages for ASCL1 and FOXO1 binding, whereas ASCL1-Dam and FOXO1-Dam highlights that these transcriptional factors indeed bind to those target gene, which represents a smaller portion of the open chromatin regions. Indeed, as pointed out by the reviewer, the lower p-values and higher enrichment values in ASCL1-Dam and FOXO1-Dam conditions compared to the Dam-alone support our conclusions. Although increased, the enrichment may seem modest in the TF-Dam conditions. However, it is important to note that the ASCL1-Dam and FOXO1-Dam data have been normalised to Dam-alone controls (detailed in Methods). While, due to this normalisation, some of the rarely bound regions may indeed be underrepresented in the transcription factor binding data, at the expense of these, we resolve only the most significantly bound loci.

The resolution of DamID data is not sufficient to identify how the co-bound target gene expression is regulated. Some potential mechanisms were mentioned in the Discussion. Finally, we would like to note that the Dam-alone analysis was also performed during the neural differentiation, where the chromatin landscape is permissive for neural differentiation, and therefore, enriched motifs for relevant transcription factors are expected. An important future experiment would be to assess chromatin accessibility prior to differentiation and in other cell types, such as astroglial cells, to test if the target genes of transcription factors supporting neural differentiation would still be accessible.

We modified our manuscript to highlight some of these points in the Discussion, within the constraints of the word limit and the scope of our manuscript.

2. The authors could mention as a limitation of the study that the distal regulatory elements are excluded from the target analyses of Ascl1 and Foxo1.

We included this in the limitations of the study.

3. In Cell2Fate analyses, it should be explained why the authors decided to focus on modules 2, 4 and 6. Furthermore, they should explain the selection of the GO terms shown in Fig 5F-H. For example, the GO terms of the module 2 (Table S6) include both positive and negative regulation of Wnt signalling. In the Fig 5F and in the text, the authors emphasize positive regulation of Wnt signalling in the module 2.

Thank you for the comment. Please see lines 543-546 for the explanation of why we focused on modules 2, 4, and 6. We included the observation regarding the GO terms enriched for module 2 genes in the relevant section. While the negative regulation of WNT is observed in both modules 2 and 6, terms associated with positive regulation are only observed in Module 2. This observation continues to suggest a temporarily restricted WNT signalling activation and a potential negative feedback at the cellular states that coincide with module 2. Further experimental validation is required to understand the precise regulation of the WNT signalling pathway in this context. We expanded the discussion on this point.

4. There are a few unclear sentences in the manuscript. For example line 164-165, line 169-170, line 267-268

We edited the problematic sentences.

Reviewer 3:

In the revised manuscript, the authors have addressed most of our previous concerns, with the exception of one point detailed below. The newly provided data have strengthened the overall quality of the paper.

We would like to thank the reviewer for their constructive suggestions.

Given the senior authors' previous publications, the substantial number of GCPs recovered in the dataset is somewhat unexpected. The response to our original Point 5 is not entirely accurate. While several studies have reported Nes⁺ cells in the EGL, none have demonstrated co-expression of Nes and *Atoh1*. Notably, Peng et al. (Nature Neuroscience, 2013) showed that Nes⁺ cells in the EGL are negative for *Atoh1*. The authors should revise the manuscript to accurately describe this Nes⁺/*Atoh1*⁺ population and clarify its novelty. The study by Peng et al. (Nature Neuroscience, 16(12):1737-44, 2013) is highly relevant and should be cited.

We thank the reviewer for their excitement about the potential importance of the Nes⁺ GCPs and for highlighting the omitted reference. We avoid conclusions based on the abundance of the cells in the scRNA-seq data, as this is heavily affected by cells' ability to survive dissociation and FACS, and likely does not reflect the physiological proportions of different cell types within the cerebellum. As can be seen in the analysis in Pakula *et al*, 2025, although *Atoh1* is significantly enriched, the levels of *Atoh1* expression are low and patchy in these clusters, and therefore some of the GCPs we obtained in our analysis may indeed be a similar population described in Peng Li *et al.*, 2013. However, such interpretations could be misleading due to the high drop-out rates of scRNA-seq analyses. We included the reference and elaborated further on these cells. However, we believe that further characterisation and validation of these observations are needed before claiming these are distinct/novel GCP populations from what was reported before. Future studies should systematically investigate the differences between different Sox2⁺ and/or Nes⁺ GCP populations to the rest of the GCPs using single-cell genomics at high resolution and thorough histological analysis and fate mapping. Given that our scRNA-seq data was generated only from the Nes-CFP⁺ cells, we don't have the resolution to compare different GCP populations that have varying levels of VZ markers and *Atoh1*. We apologise for not being able to discuss this point further due to the word limit and the context of this manuscript.

Third decision letter

MS ID#: dev.204811R2

MS Title: A conserved differentiation program facilitates inhibitory neuron production in the developing mouse and human cerebellum

Authors: Jens Bager Christensen; Alex P. A. Donovan; Marzieh Moradi; Giada Vanacore; Mohab Helmy; Adam J. Reid; Jimmy Tsz Hang Lee; Omer Ali Bayraktar; Andrea H. Brand; N. Sumru Bayin
Article Type: Research Article

Dear Dr Bayin,

I am delighted to let you know that your manuscript has been accepted for publication in Development, pending our standard publication integrity checks.

Reviewer 1

The authors have appropriately addressed most of my comments. They have also provided thoughtful discussion regarding the limitations of the in vitro neurosphere culture and DamID experimental systems. With the addition of new in vivo data, the experiments that are feasible at this stage have been conducted, and the revised manuscript appears to be substantially improved. I believe it will provide valuable insights to the field of cerebellar development research.

Reviewer 2

The authors have addressed all the points I raised in my earlier review.